# SinPoint: A Novel Topological Consistent Augmentation for Point Cloud Understanding

## Abstract

Data augmentation is a highly effective method for addressing the issue of data scarcity in machine learning and computer vision tasks. It involves diversifying the original data through a series of transformations to improve the robustness and generalization ability of the model. However, due to the disorder and irregularity of point clouds, existing methods struggle to enrich geometric diversity and maintain topological consistency, leading to imprecise point cloud understanding. In this paper, we propose SinPoint, a novel method designed to preserve the topological structure of the original point cloud through a homeomorphism. Additionally, it utilizes the Sine function to generate smooth displacements. This simulates object deformations, thereby producing a rich diversity of samples. Our extensive experiments demonstrate that SinPoint consistently outperforms existing Mixup and Deformation methods on various benchmark point cloud datasets, improving performance for shape classification and part segmentation tasks. Specifically, when used with PointNet++ and DGCNN, SinPoint achieves a state-of-the-art accuracy of 90.2 on shape classification with the real-world ScanObjectNN dataset. Furthermore, our method is highly versatile and scalable, and it can adapt to different scenarios and requirements for point cloud tasks.

## 1 Introduction

With the widespread application of deep learning in computer vision, the point cloud, as a crucial representation of three-dimensional shapes, is gradually attracting attention from the industry Qi et al. (2017a;b); Li et al. (2018); Thomas et al. (2019); Lin et al. (2020a); Qian et al. (2022). Compared to traditional two-dimensional images, point clouds offer richer three-dimensional spatial information, providing broader application prospects for areas such as autonomous driving Navarro-Serment et al. (2010); Kidono et al. (2011); Chen et al. (2017), scene understanding Verdoja et al. (2017); Chen et al. (2019a); Sugimura et al. (2020), and virtual reality Rusu et al. (2008); Qi et al. (2018). However, due to the dual constraints of technology and cost, there are relatively small-scale 3D datasets with limited labels and diversity. This limitation can cause the problem of overfitting and further affect the generalization ability of the network. Facing such challenges, effectively using limited point cloud data for better training deep learning models has been an urgent problem to address. Recently, **Data Augmentation** (DA) has become a popular strategy in avoiding overfitting and improving the generalizability of models by increasing the quantity and diversity of samples.

DA has shown significant success in handling image data. Various methods, such as Cutout De-Vries & Taylor (2017); Zhong et al. (2020), Mixup Zhang et al. (2017), Cutmix Yun et al. (2019), and other methods Verma et al. (2019); Yang et al. (2022), have been utilized to augment images and enhance the robustness and generalizability of models. However, unlike regular images, point clouds are disordered and irregular, making it challenging to apply these DA methods directly to point cloud data. Some existing point cloud DA methods only focus on a single type of operation, such as simple geometric transformations (rotation, scaling, and translation), data perturbations (adding noise and deleting points), or hybrid operations (simulated mixtures of images Chen et al. (2020); Lee et al. (2022; 2021); Zhang et al. (2022); Ren et al. (2022); Wang et al. (2024)). While these methods may improve data diversity to some extent, they often overlook the point cloud's intrinsic structure and semantic details, resulting in a loss of topological consistency in the augmented point cloud. For instance, PointMixup Chen et al. (2020), Point-CutMix Zhang et al. (2022), and SageMix Lee et al. (2022) all use different strategies to mix

samples, but they do not consider the local structure of each sample. PointAugment Li et al. (2020) relies on a learnable transformation matrix, making the outcome unpredictable. Similarly, PointWOLF Kim et al. (2021) transforms local point clouds using a combination of strategies, which can lead to data distortion and significant semantic deviation, as shown in Figure 1.

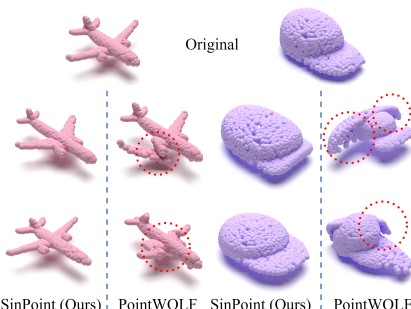

In point cloud processing, maintaining topological consistency is crucial for ensuring the accuracy and performance of the model. Studies have shown that frameworks such as PointNet++ Qi et al. (2017b), DGCNN Wang et al. (2019b), Point Cloud Transformer Guo et al. (2021), and PointMLP Ma et al. (2022) can improve the embedded representation of input data by using techniques like farthest point sampling and k-nearest neighbor search. These frameworks effectively capture the spatial structure and contextual information of the point cloud by preserving the point cloud's topological relationships. Further investigation Lee et al. (2022) has revealed that when the topology of the point cloud is destroyed, the model cannot learn the correct representation, leading to decreased performance. Therefore, in point cloud augmentation, it is vital to maintain topological consistency while increasing the geometric diversity of samples. However, unlike topological consistency, which emphasizes keeping the connections between points unchanged, geometric diversity usually refers to the augmented point cloud having a wider range of shapes, sizes, and spatial distributions than the original point cloud. In other words, geometric diversity is the representation, while topological consistency is the essence. These two requirements can help ensure the validity and reliability of the augmented point cloud.

Figure 1: SinPoint ensures a smooth and natural deformation process, avoiding abrupt or unnatural changes. Point-WOLF can lead to data distortion and significant semantic deviation.

This paper proposes a novel SinPoint transformation technique based on a homeomorphism Derrick (1973) to address these issues mentioned above. SinPoint aims to preserve the topological structure of the original point cloud by a homeomorphism Derrick (1973) and perturb the local structure using a Sine function to simulate the deformation of objects, thereby expanding the diversity of point clouds. We design two deformation strategies, as shown in Figure 2. One is to use a Single Sine Function (SinPoint-SSF) with the initial phase as the origin to deform the point cloud. The other is to use Multiple Sine Function (SinPoint-MSF), with different anchor points as the initial phase. The sine transforms of different parameters are superimposed to obtain richer deformations. We experimentally demonstrate that SinPoint outperforms the State-Of-The-Art (SOTA) point cloud augmentation method on multiple datasets.

In summary, our contribution has three main aspects: 1) We propose a novel method for point cloud DA from the viewpoint of topological consistency and geometric diversity. To the best of our knowledge, this is the first work that incorporates a homeomorphism and Sine function in point cloud DA. 2) We prove that the proposed Sine-based mapping function is a homeomorphism. This can make the local region of the point cloud produce deformed shapes without destroying the topology, thereby enhancing the diversity of the point cloud. 3) We demonstrate the effectiveness of our framework by showing consistent improvements over state-of-the-art augmentation methods on both synthetic and real-world datasets in 3D shape classification and part segmentation tasks. Our extensive experiments show that SinPoint improves the generalization and robustness of various models.

## 2 RELATED WORK

**Deep learning on point cloud.** PointNet Qi et al. (2017a) is a pioneering work that uses shared MLPs to encode each point individually and aggregates all point features through global pooling. Inspired by CNNs, PointNet++ Qi et al. (2017b) adopts a hierarchical multi-scale or weighted feature aggregation scheme to get local features. DGCNN Wang et al. (2019b) introduces EdgeConv, which utilizes edge features from the dynamically updated graph. Additionally, various works have focused on point-wise multi-layer perceptron Liu et al. (2020); Xu et al. (2021c); Shen et al. (2018);

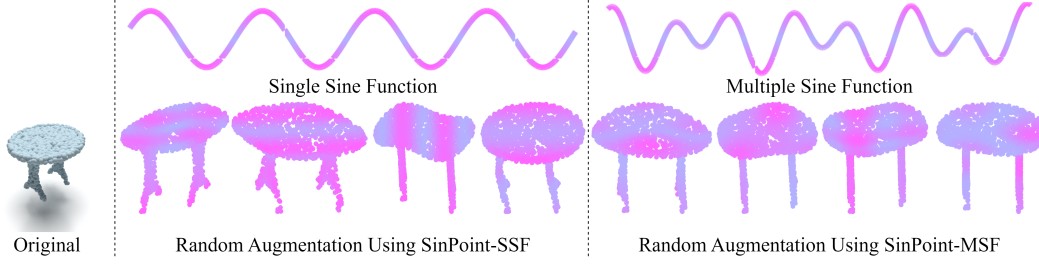

Figure 2: Some augmented samples produced by SinPoint-SSF (left) and SinPoint-MSF (right). The color is close to pink, the deformation is more significant. The data generated by the SinPoint-SSF object has only a single peak, while the samples from SinPoint-MSF have multiple distinct peaks.

Ma et al. (2022); Qian et al. (2022); Zhang et al. (2023), convolution Li et al. (2018); Wu et al. (2019); Thomas et al. (2019); Lin et al. (2020a); Xu et al. (2021b); Liu et al. (2019) , and graph-based Qi et al. (2017c); Simonovsky & Komodakis (2017); Wang et al. (2019a); Lin et al. (2020b; 2021) methods to process point clouds. These methods consistently use Conventional Data Augmentation (CDA) Qi et al. (2017a;b); Wang et al. (2019b) to improve the model's robust kernel generalization performance, but the improvement is relatively marginal. Parallel to these approaches, other recent works Chen et al. (2020); Li et al. (2020); Lee et al. (2022; 2021); Zhang et al. (2022); Kim et al. (2021); Ren et al. (2022); Hong et al. (2023) focus on data augmentation to improve the generalization power of deep neural networks in point clouds.

**Data augmentation on point cloud.** Current point cloud augmentation methods can be divided into two categories: self-augmentation and mix-augmentation. Self-augmentation through geometric transformation to augment the shape diversity of the point cloud. For instance, CDA Qi et al. (2017a;b); Wang et al. (2019b) encompasses geometric transformations like rotation, scaling, translation, and jittering, alongside the addition of noise and point reduction to enhance sample diversity. PointAugment Li et al. (2020) learn the transformation matrix with an augmentor network to produce augmentations. PointWOLF Kim et al. (2021) selects various anchor points to serve as central points for the local point cloud's weighted transformation, leading to smooth and varied non-rigid deformations. Mix-augmentation uses different strategies to cut and combine two point clouds to form a new point cloud that contains two local shapes. For example, PointMixup Chen et al. (2020) recently used the shortest path linear interpolation between instances to augment data in the point cloud. PointCutMix Zhang et al. (2022) benefits from CutMixup and PointMixup, and proposes cutting and pasting of point cloud parts. SageMix Lee et al. (2022) proposes a saliency-guided Mixup for point clouds to preserve salient local structures.

## 3 METHOD

### 3.1 PRELIMINARIES

Homeomorphism Derrick (1973) is an important mathematical tool used to describe the equivalence relations between topological spaces. It keeps the proximity of points in space unchanged and makes topological Spaces have the same topological properties. These properties make a homeomorphism widely used in topology, geometry, physics, and other fields.

**Definition 1. (Homeomorphism)** Given two topological spaces $X, Y$, and given a mapping $f : X \rightarrow Y$. $f$ is a homeomorphism of two spaces when it is satisfied that $f$ is a bijection and $f$ and $f^{-1}$ are continuous, denoted as $X \cong Y$.

**Definition 2. (Local homeomorphism)** Let $f : X \rightarrow Y$ is a mapping between two topological spaces $X$ and $Y$. If for every point $x$ in $X$, exists a neighborhood $U$ of $x$ such that $f(U)$ is an open set in $X$ and $f_U : U \rightarrow f(U)$ is a homeomorphism, then $f$ is a local homeomorphism.

They have similar mathematical properties, both of which require the mapping and its inverse mapping to be continuous and maintain the topological structure of the spaceArmstrong (2013).

**Proposition 1. (Topological consistency)** If $f$ is a homeomorphism from $X$ to $Y$, then $X$ and $Y$ have the same topological properties.

**Proposition 2. (Reflexivity, symmetry, and transitivity)** The homeomorphic relation is an equivalence relation, and therefore it has reflexivity (any topological space is a homeomorphism to itself), symmetry (if $X \cong Y$, then $Y \cong X$), and transitivity (if $Y \cong Z$, then $X \cong Z$).

Homeomorphism plays a vital role in ensuring topological consistency. First, a homeomorphism is not only one-to-one (bijective) and continuous, but its inverse is also continuous. This guarantees the preservation of the space topology. Second, when processing point clouds, using a homeomorphism can ensure that the basic shape and structure of the point cloud remain unaltered after enhancement or transformation, thus making the augmented data more consistent with the actual scene or object.

### 3.2 RESIDUAL FUNCTION

Inspired by the deep residual network He et al. (2016), we focus on obtaining continuous residual coordinates and generating augmented coordinates by adding offsets to the original coordinates. Specifically, for a given point cloud $P = \{p_1, p_2, ..., p_n\}$, we only need to compute the residual coordinates, represented by $P' - P$, thus the augmentation process becomes:

$$P' = P + g(P). \tag{1}$$

A homeomorphism can ensure that the original space and the deformed space have topological consistency. Therefore, it is very important to select the appropriate residual function, which not only gains diversity but also needs to guarantee that the whole mapping is homeomorphic.

### 3.3 SINPOINT

To simulate the distortion and deformation of an object, we have chosen to use the Sine function as our residual function. The inherent periodic nature of the Sine function allows us to adjust the number of regions that are deformed with precision. Additionally, by manipulating the amplitude of the Sine function, we can precisely control the intensity of the deformation. This displacement field, generated by the Sine function, effectively distorts and deforms specific local regions of the point cloud data without altering the overall topology. As a result, the augmented point cloud data contains more intricate and detailed local features. The standard Sine function is shown below:

$$g(x) = A\sin(\omega x + \varphi), \tag{2}$$

where $A$, $\omega$, and $\varphi$ represent the amplitude, the angular velocity (control period), and the initial phase, respectively. The displacement field generated by Sine function is introduced into a homeomorphism, and a homeomorphism based on Sine function can be obtained.

**Theorem 1. (Homeomorphism Based on Sine Function)** Given two topological spaces $X, Y$, and given a mapping $f : X \rightarrow Y = X + A\sin(\omega X + \varphi)$, if $-1 \leq A\omega \leq 1$, then $f$ is a homeomorphism, else $f$ is a local homeomorphism. (**The proof is in the Appendix.**)

Since $f$ is a homeomorphism, we can use it to augment the point cloud and ensure its topological consistency. Given a set of points $P = \{p_1, p_2, ..., p_n\}$, where $N$ represents the number of points in the Euclidean space $(x, y, z)$. SinPoint applies a homeomorphism and the resulting augmented point cloud $P'$ is given as follows:

$$P' = P + A\sin(\omega P + \varphi), \tag{3}$$

where $A\sin(\omega P + \varphi)$ is displacement field of $P$. We need to adjust $A$ and $\omega$ to produce more diverse point clouds. In this paper, we set $A \sim U(-a, a)$ and $\omega \sim U(-w, w)$ to obey the uniform distribution. In this way, more samples with smooth deformation can be generated, which makes the distribution of samples more uniform.

As illustrated in Figure 3, the transformation of the circle by Equation (3) results in continuous local indentations on the point cloud surface, attributable to the Sine function's periodic nature. This deformation technique enables the simulation of a concavity similar to that observed when an object is indented while preserving the point cloud's topology structure.

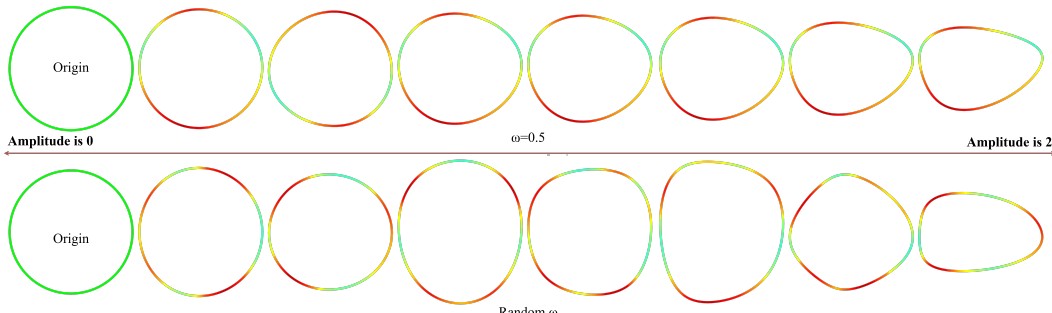

Figure 3: SinPoint obtains different degrees of geometric deformation by controlling the amplitude and angular velocity $\omega$. In the above, only two peaks or troughs appear due to the fixed angular velocity $\omega$. In the following, multiple peaks or troughs occur due to the random angular velocity $\omega$. The darker the color, the larger the deformation.

We designed two transformation strategies for SinPoint, SinPoint-SSF based on a single sine function and SinPoint-MSF based on the superposition of multiple sine functions. Algorithm 1 is the process of generating enhanced samples. Different strategies generate various deformation samples. The two detailed strategies are as follows:

---

**Algorithm 1** SinPoint

---

**Input:** Original point cloud $P = \{p_1, p_2, ..., p_n\}$
         Condition $key$ for SinPoint-SSF or SinPoint-MSF
         Anchor points number $k$, Amplitude $a$, Angular velocity $\omega$
**Output:** $P^{'}$
  1: **if** $key ==$ "SSF"
  2:    Sample $A \sim U(-a, a)$
  3:    Sample $\omega \sim U(-w, w)$
  4:    $P^{'} \leftarrow P + A sin(\omega P)$
  5: **else if** $key ==$ "MSF"
  6:    **if** using farthest point sampling
  7:      $P_i \leftarrow FPS(P, k)$, *#FPS() is farthest point sampling*
  8:    **else if** using random point sampling
  9:      $P_i \leftarrow RPS(P, k)$, *#RPS() is random point sampling*
10:    **end if**
11:    Sample $A_i \sim U(-a, a), i = 1 : k$
12:    Sample $\omega_i \sim U(-w, w), i = 1 : k$
13:    $P^{'} \leftarrow P + \frac{1}{k} \sum_{i=1}^{k} \{A_i sin(\omega_i P + P_i)\}$
14: **end if**
15: **Return** $P^{'}$

---

**SinPoint-SSF** uses a single sine function to perturb the coordinates of the point cloud. We normalize the point cloud to the unit sphere space and take the sphere's center as the initial phase, that is, $\varphi = 0$. Then the SinPoint-SSF transformed point cloud can be expressed as follows:

$$P^{'} = P + A sin(\omega P). \tag{4}$$

**SinPoint-MSF** superposes multiple sine functions to perturb the point cloud. Multiple sinusoidal complex waves exhibit rich waveform characteristics through diverse combinations of frequency, amplitude and phase. SinPoint-MSF first selects $k$ anchor points as the initial phase and samples different amplitudes and angular velocities for additive perturbations. This provides more diversity to the point cloud and generates realistic samples. The transformation of SinPoint-MSF as follows:

$$P^{'} = P + \frac{1}{k} \sum_{i=1}^{k} \{A_i sin(\omega_i P + \varphi_i)\}, \ \varphi_i = P_i. \tag{5}$$

Figure 4: An overview of our SinPoint framework. We use SinPoint to get the augmented point cloud and input it into the network with the original point cloud for training. SinPoint can be adapted for a variety of tasks due to label consistency and topological consistency.

Our framework is shown in Figure 4. It maps the point cloud input to a feature space with topological consistency. By incorporating augmented inputs generated by SinPoint, the training process optimizes the backbone's parameters in a larger feature space than the baseline. During the training stage, by utilizing the same loss function as the baseline and optimizing in this new augmented feature space, our method learns a more expressive representation that better fits the input data, leading to improved generalization. During inference, the model uses the original point cloud input to preserve the data's geometric structure, ensuring no disruption to geometric priors in practical applications. Our method maintains point cloud geometric and topological properties. It also extracts more discriminative features, significantly enhancing overall model performance. Our method has been validated on object point clouds.

## 4 EXPERIMENTS

In this section, we demonstrate the effectiveness of our proposed method, SinPoint, with various benchmark datasets and baselines. First, for 3D shape classification, we evaluate the generalization performance and robustness using SinPoint-SSF in Section 4.1. Next, we compare our SinPoint-MSF with existing data augmentation methods in Section 4.2 for part segmentation. More ablation studies and implementation details are provided in **Appendix**.

**Datasets.** For classification task, we use two **synthetic datasets**: ModelNet40 (**MN40**) Wu et al. (2015) and Reduced MN40 (**RMN40**), and two **real-world datasets** from ScanObjectNN Uy et al. (2019): **OBJ_ONLY** and **PB_T50_RS**. MN40 is a widely used synthetic benchmark dataset containing 9840 CAD models in the training set and 2468 CAD models in the validation set, with a total 40 classes of common object categories. RMN40 comes from PointMixup Chen et al. (2020) and only contains 20% training samples to simulate data scarcity. ScanObjectNN is a real-world dataset that is split into 80% for training and 20% for evaluation. Among the variants of ScanObjectNN, we adopt the simplest version (OBJ_ONLY) and the most challenging version (PB_T50_RS). OBJ_ONLY, which has 2,309 and 581 scanned objects for the training and validation sets, respectively, and PB_T50_RS, which is a perturbed version with 11,416 and 2,882 scanned objects for the training and validation sets, respectively. Both have 15 classes. We use only coordinates $(x, y, z)$ of 1024 points for training models without additional information, such as the normal vector. For the part segmentation task, we adopt a synthetic dataset, **ShapeNetPart** Yi et al. (2016), which contains 14,007 and 2,874 samples for training and validation sets. ShapeNetPart consists of 16 classes with 50 part labels. Each class has 2 to 6 parts.

**Baselines.** For comparison with previous studies, we use three backbone models: PointNet Qi et al. (2017a), PointNet++ Qi et al. (2017b), and DGCNN Wang et al. (2019b). We compare SinPoint with the model under default augmentation in Base Qi et al. (2017a;b); Wang et al. (2019b) and eight SOTA methods. We report the performance in terms of overall accuracy. To further verify the effectiveness of SinPoint, we add a variety of backbone networks, including RSCNN Liu et al. (2019), PointConv Wu et al. (2019), PointCNN Li et al. (2018), GDANet Xu et al. (2021c), PCT Guo et al. (2021) and PointMLP Ma et al. (2022).

Table 1: 3D shape classification performance on ModelNet40 /ScanObjectNN

| Model | Method | ModelNet40 | | ScanObjectNN | |
|---|---|---|---|---|---|
| | | MN40 | ReducedMN40 | OBJ_ONLY | PB_T50_RS |
| PointNet | Base | 89.2 | 81.9 | 76.1 | 64.1 |
| | +PointMixup | 89.9 | 83.4 | - | - |
| | +RSMix | 88.7 | - | - | - |
| | +SageMix | 90.3 | - | 79.5 | 66.1 |
| | +PointCutMix | 90.5 | - | - | - |
| | +PointAugment | 90.9 | 84.1 | 74.4 | 57.0 |
| | +PatchAugment | 90.9 | - | - | - |
| | +PointWOLF | 91.1 | 85.7 | 78.7 | 67.1 |
| | +WOLFMix | 90.7 | - | - | - |
| | +PCSalMix | 90.5 | - | - | - |
| | +PointPatchMix | 90.1 | - | - | - |
| | **+SinPoint(Ours)** | **91.3 (↑ 2.1)** | **86.5 (↑ 4.6)** | **82.6 (↑ 6.5)** | **70.8 (↑ 6.7)** |
| PointNet++ | Base | 90.7 | 85.9 | 84.3 | 79.4 |
| | +PointMixup | 92.3 | 88.6 | 88.5 | 80.6 |
| | +RSMix | 91.6 | - | - | - |
| | +SageMix | 93.3 | - | 88.7 | 83.7 |
| | +PointCutMix | 93.4 | - | - | - |
| | +PointAugment | 92.9 | 87.0 | 85.4 | 77.9 |
| | +PatchAugment | 92.4 | - | 87.1 | 81.0 |
| | +PointWOLF | 93.2 | 88.7 | 89.7 | 84.1 |
| | +WOLFMix | 93.1 | - | - | - |
| | +PCSalMix | 93.1 | - | - | - |
| | +PointPatchMix | 92.9 | - | - | - |
| | **+SinPoint(Ours)** | **93.4 (↑ 2.7)** | **89.6 (↑ 3.7)** | **90.2 (↑ 5.9)** | **84.5 (↑ 5.1)** |
| DGCNN | Base | 92.2 | 87.5 | 86.2 | 77.3 |
| | +PointMixup | 92.9 | 89.0 | - | - |
| | +RSMix | 93.5 | - | - | - |
| | +SageMix | 93.6 | - | 88.0 | 83.6 |
| | +PointCutMix | 93.2 | - | - | - |
| | +PointAugment | 93.4 | 88.3 | 83.1 | 76.8 |
| | +PatchAugment | 93.1 | - | 86.9 | 79.1 |
| | +PointWOLF | 93.2 | 89.3 | 88.8 | 81.6 |
| | +WOLFMix | 93.2 | - | - | - |
| | +PCSalMix | 93.2 | - | - | - |
| | **+SinPoint(Ours)** | **93.7 (↑ 1.5)** | **90.1 (↑ 2.6)** | **90.2 (↑ 4.0)** | **84.6 (↑ 7.3)** |

Table 2: 3D shape classification performance in various architectures on ModelNet40.

| Model | PointNet | PointNet++ | DGCNN | RSCNN | PointConv | PointCNN | GDANet | PCT | PointMLP |
|---|---|---|---|---|---|---|---|---|---|
| Param. | - | 1.4M | 1.8M | - | 18.6M | - | - | 2.8M | 12.6M |
| Base | 89.2 | 90.7 | 92.2 | 91.7 | 92.5 | 92.5 | 93.4 | 93.2 | 94.1 |
| **+SinPoint** | **91.3 (↑ 2.1)** | **93.4 (↑ 2.7)** | **93.7 (↑ 1.5)** | **92.9 (↑ 1.2)** | **92.8 (↑ 0.3)** | **93.2 (↑ 0.7)** | **93.6 (↑ 0.2)** | **93.5 (↑ 0.3)** | **94.3 (↑ 0.2)** |

## 4.1 3D SHAPE CLASSIFICATION

**Comparisons with SOTA Methods**. Experimental results of 3D shape classification are shown in Table 1. We report the Overall Accuracy (OA) of each model on all four datasets. From the results, we can clearly see that our SinPoint significantly outperforms all of the previous methods in every dataset and model. Particularly, the average OA improvement on the synthetic datasets is **2.6%**, and the average OA improvement on the real-world datasets is even **5.9%**, and the maximum improvement was DGCNN reaching **7.3%** in PB_T50_RS. It proves that our SinPoint is more efficient on real data sets. These consistent improvements demonstrate the effectiveness of our framework.

**3D shape classification performance under Various Network Backbones.** The effectiveness of SinPoint is further validated across a variety of network architectures in ModelNet40 Wu et al. (2015) and ScanObjectNN Uy et al. (2019), including PointNet Qi et al. (2017a), PointNet++ Qi

et al. (2017b), DGCNN Wang et al. (2019b), RSCNN Liu et al. (2019), PointConv Wu et al. (2019), PointCNN Li et al. (2018), GDANet Xu et al. (2021c), PCT Guo et al. (2021) and PointMLP Ma et al. (2022), PointNeXt-S Qian et al. (2022), PointMetaBase-S Lin et al. (2023), SPoTr Park et al. (2023). From Table 2, we observe that SinPoint has a consistent improvement of accuracy against the baselines (+0.2∼2.7%). Notably, using the basic PointNet++ and DGCNN, we can surpass the Transformer-based baseline PCT while reducing the parameters by half. This reduction compensates for the parameter deficiency in the network through data augmentation alone. Surprisingly, DGCNN+SinPoint is only 0.4% lower than PointMLP, but the parameters are 7 times lower. Table 3 compares some of the latest backbone networks.

Table 3: 3D shape classification performance in various architectures on PB_T50_RS

| Model | PointMLP ICLR 2022 | PointNeXt-S NeurIPS 2022 | PointMetaBase-S CVPR 2023 | SPoTr CVPR 2023 |
|---|---|---|---|---|
| Base | 85.7 | 87.7 | 88.2 | 88.6 |
| **+SinPoint(Ours)** | **87.5(+1.8)** | **88.9(+1.2)** | **89.3(+1.1)** | **89.5(+0.9)** |

Table 4: Robustness with DGCNN Wang et al. (2019b) on OBJ_ONLY Uy et al. (2019)

| Method | Gaussian noise | | Rotation 180$^o$ | | Scaling | | Dropout | |
|---|---|---|---|---|---|---|---|---|
| | $\sigma$:0.01 | $\sigma$:0.05 | X-axis | Z-axis | ×0.6 | ×2.0 | 25% | 50% |
| DGCNN | 84.9 | 48.4 | 32.5 | 32.4 | 73.7 | 73.0 | 83.3 | 75.7 |
| + PointMixup | 85.0 | 61.3 | 31.7 | 32.7 | 73.8 | 73.0 | 84.2 | 74.9 |
| + RSMix | 84.2 | 49.1 | 32.7 | 32.6 | 75.0 | 74.5 | 84.0 | 73.6 |
| + SageMix | 85.7 | 51.2 | 36.5 | 37.9 | 75.6 | 75.2 | 84.9 | 79.0 |
| **+ SinPoint** | **85.9** | **61.5** | **38.6** | **44.1** | **76.1** | **75.6** | **85.1** | **79.5** |

**Robustness.** Additional studies demonstrate our SinPoint improves the robustness of models against previous methods Chen et al. (2020); Lee et al. (2021; 2022) on four types of corruption: (1) **Gaussian noise** with ($\sigma \in (0.01, 0.05)$, (2) **Rotation** $180^o$ (X-axis,Z-axis), (3) **Scaling** with a factor in 0.6, 2.0, and (4) **Dropout** with a rate $r \in \{0.25, 0.50\}$. We adopt DGCNN and OBJ_ONLY to evaluate the robustness of models. As shown in Table 4, SinPoint consistently improves robustness in various corruptions. DGCNN with SinPoint shows the best robustness with significant gains compared to previous methods. Importantly, the gain over the baseline significantly increases as the amount of corruption increases: 13.1% for Gaussian noise ($\sigma$ : 0.05), 11.7% for Rotation 180° (Z-axis), 2.6% for Scaling in 2.0, and 3.8% for Dropout (r = 0.5). We believe that the diverse samples augmented by a homeomorphism in SinPoint help models to learn more robust features against both 'local' and 'global' corruptions.

Table 5: Ablation study of SinPoint on ModelNet40 Wu et al. (2015). Mix: mixed training samples.

| DGCNN | CDA | Drop | SinPoint(Ours) | Mix(Ours) | OA | Inc.↑ |
|---|---|---|---|---|---|---|
| A | | | | | 91.7 | - |
| B | ✓ | | | | 92.2 | 0.5 |
| C | ✓ | ✓ | | | 92.7 | 1.0 |
| D | | | ✓ | | **92.9** | **1.2** |
| E | | | ✓ | ✓ | **93.2** | **1.5** |
| F | ✓ | | ✓ | ✓ | **93.4** | **1.7** |
| G | ✓ | ✓ | ✓ | ✓ | **93.7** | **2.0** |

**Ablation study of modules.** Table 5 summarizes the results of the ablation study on DGCNN. Model A gives a baseline classification accuracy of 91.7%. On top of Model A, we use a combination of different augmentors. From the results shown in Table 5, we can see that each augmentation function contributes to producing more effective augmented samples. It is worth noting that when only SinPoint is used, the results already surpass A, B, and C, while using a mixture of original and augmented samples can again improve the generalization performance of DGCNN. Moreover, when

using more modules, the model's generalization ability is further improved to 93.7%, an improvement of 2.0% over the vanilla.

## 4.2 3D Shape Part Segmentation

SinPoint can also be generalized to other 3D point cloud tasks due to its geometric consistency and label consistency. We first construct SinPoint-SSF and SinPoint-MSF following the optimal parameter settings in the classification task. Next, we test SinPoint for 3D shape part segmentation task on the ShapeNetPart Yi et al. (2016) benchmark. We follow the settings from PointNet, PointNet++ and DGCNN that randomly select 2048 points as input for a fair comparison. We compare our methods with several recent methods, including PointMixup Chen et al. (2020), RSMix Lee et al. (2021), SageMix Lee et al. (2022) and PointWOLF Kim et al. (2021). Table 7 shows that on ShapeNetPart, SinPoint-MSF consistently improves mean IoU (mIoU) over baselines (0.9% over PointNet, 1.0% over PointNet++ and 0.7% over DGCNN), demonstrating the applicability of SinPoint-MSF to point-wise tasks. Notably, SinPoint-MSF has optimal performance in most shapes. It also proves that SinPoint-MSF with more local deformations is effective in part segmentation. Furthermore, Table 6 presents comparative results with state-of-the-art methods, and in the **Appendix**, we offer comparisons against a variety baseline models, where our SinPoint consistently outperforms others. Finally, in Figure 5, we present the visualization results for Sinpoint and baseline.

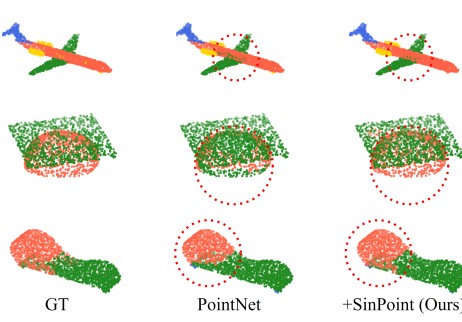

GT      PointNet      +SinPoint (Ours)

Figure 5: 3D part segmentation experiment visualization results.

Table 6: Overall mean Inter-over-Union (mIoU) on ShapeNetPart Yi et al. (2016).

| Method | mIoU | aero | bag | cap | car | chair | aer | guitar | knife | lamp | laptop | motor | mug | pistol | rocket | skate | table |
|---|---|---|---|---|---|---|---|---|---|---|---|---|---|---|---|---|---|
| PointNet | 83.5 | 81.8 | 74.7 | 80.2 | 71.9 | 89.6 | 71.5 | 90.3 | 84.9 | 79.5 | 95.2 | 65.2 | 91.1 | 81.1 | 55.1 | 72.8 | 82.2 |
| +PointWOLF | 83.8 | 82.5 | 73.3 | 78.8 | 73.2 | 89.6 | **72.2** | 91.2 | 86.2 | **79.7** | 95.2 | 64.6 | 92.5 | 80.2 | **56.6** | 73.1 | 82.2 |
| **+SinPoint-SSF** | 84.3 | **82.6** | 79.8 | 80.5 | **77.9** | 89.4 | 70.2 | 91.4 | 86.6 | 79.2 | 95.6 | 67.6 | **93.9** | 82.7 | 54.6 | 80.5 | 82.7 |
| **+SinPoint-MSF** | **84.4** | 82.4 | **81.0** | **84.1** | 77.3 | **89.6** | 70.1 | **91.4** | **86.9** | 79.4 | **95.6** | 67.6 | 93.6 | **82.9** | 54.5 | **81.5** | **82.8** |
| PointNet++ | 84.8 | 81.9 | 83.4 | 86.4 | 78.6 | 90.5 | 64.7 | **91.4** | 83.1 | 83.4 | 95.1 | 69.6 | 94.7 | **82.8** | 56.9 | 76.0 | 82.3 |
| +PointWOLF | 85.2 | 82.0 | **83.9** | 87.3 | 77.6 | 90.6 | 78.4 | 91.1 | 87.6 | 84.7 | 95.2 | 62.0 | 94.5 | 81.3 | 62.5 | 75.7 | 83.2 |
| **+SinPoint-SSF** | 85.7 | **83.2** | 82.3 | **89.0** | 79.2 | 91.0 | **81.1** | 91.2 | **88.4** | 84.2 | **95.9** | 70.2 | **95.6** | 82.3 | **62.8** | 75.5 | 83.1 |
| **+SinPoint-MSF** | **85.8** | 83.1 | 80.1 | 87.3 | 79.1 | **91.1** | 77.4 | 91.1 | 88.1 | **85.1** | 95.6 | **72.8** | 95.4 | 81.8 | 60.7 | **75.8** | **83.4** |
| DGCNN | 84.8 | 82.2 | 75.1 | 81.3 | 78.2 | 90.6 | 73.6 | 90.8 | 87.8 | 84.4 | 95.6 | 57.8 | 92.8 | 80.6 | 51.5 | 73.9 | 82.8 |
| +PointWOLF | 85.2 | 82.9 | 73.3 | 83.5 | 76.7 | **90.8** | 76.7 | **91.4** | **89.2** | 85.2 | **95.8** | 53.7 | 94.0 | 80.1 | 54.9 | 74.3 | 83.4 |
| **+SinPoint-SSF** | 85.3 | 82.5 | 84.7 | 86.3 | 77.5 | 90.5 | **76.8** | 91.0 | 88.5 | 85.2 | 95.0 | 61.0 | **94.8** | **83.1** | **62.6** | 74.9 | **83.2** |
| **+SinPoint-MSF** | **85.5** | **83.1** | **86.3** | **87.0** | **78.9** | 90.7 | 74.3 | 91.2 | 87.8 | **85.9** | 95.5 | **62.6** | 94.6 | 82.0 | 61.3 | 74.0 | 83.1 |

Table 7: Complete part segmentation results (mIoU) on ShapeNetPart Yi et al. (2016).

| Model | Base | +CDA | +PointMixup | +RSMix | +SageMix | +PointWOLF | **+SinPoint(Ours)** |
|---|---|---|---|---|---|---|---|
| PointNet++ | 84.8 | 85.1 | 85.5 | 85.4 | 85.7 | 85.2 | **85.8 (↑ 1.0)** |
| DGCNN | 84.8 | 85.0 | 85.3 | 85.2 | 85.4 | 85.2 | **85.5 (↑ 0.7)** |

## 5 Conclusion

We propose SinPoint, a novel data augmentation based a homeomorphism for Point Clouds that deforms point clouds by Sine functions while maintaining topological consistency. We only need to adjust the amplitude and frequency of the Sine function to generate a diverse and realistic sample with smoothly varying deformations, which brings significant improvements to point cloud tasks across several datasets. In the end, we conducted extensive experiments and demonstrated how Sin-Point improves the performance of three representative networks on multiple datasets. Our findings show that the augmentations we produce are visually realistic and beneficial to the models, further validating the importance of our approach to understanding the local structure of point clouds.

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

## A    APPENDIX

In this Appendix, we provide detailed discussions and experimental results. They include: 1) Implementation details of SinPoint, 2) Proof of Theorem 1: a homeomorphism Based on Sine Function; and 3) Additional ablation studies and analyses.

## B    A IMPLEMENTATION DETAIL

We conduct experiments using Python and PyTorch with two NVIDIA TITAN RTX for point clouds. Following the original configuration in Qi et al. (2017a;b); Wang et al. (2019b), we use the SGD optimizer with an initial learning rate of $10^{-1}$ and weight decay of $10^{-3}$ for PointNet Qi et al. (2017a) and PointNet++ Qi et al. (2017b) and SGD with an initial learning rate of $10^{-2}$ and weight decay of $10^{-4}$ for DGCNN Wang et al. (2019b). We train models with a batch size of 32 for 300 epochs. For a fair comparison with previous works Chen et al. (2020); Zhang et al. (2022), we also adopt conventional data augmentations with our framework (i.e., scaling and shifting for MN40 Wu et al. (2015) and rotation and jittering for ScanObjectNN Uy et al. (2019)). For hyperparameters of SinPoint-SSF and SinPoint-MSF, we opt $A = 0.6, w = 2.5, k = 4$ in the entire experiment.

## C    PROOF

**Theorem 1. (Homeomorphism Based on Sine Function)** Given two topological spaces $X, Y$, and given a mapping $f : X {\rightarrow} Y = X + Asin(\omega X + \varphi)$, if $-1 {\leq} A\omega {\leq} 1$, then $f$ is a homeomorphism, else $f$ is a local homeomorphism.

**1) Proof: if $-1 {\leq} A\omega {\leq} 1$, then $f$ is a homeomorphism**

Step 1: *Continuous*

Let $g(x) = x$, $h(x) = Asin(\omega x + \varphi), x \in R$. Since $g(x) = x$ is a continuous function and $h(x) = Asin(\omega x + \varphi)$ is also a continuous function, then $f(x) = g(x) + h(x) = x + Asin(\omega x + \varphi)$ must be a continuous function.

Step 2: *Bijective*

Let $f(x) = x + Asin(\omega x + \varphi), x \in R$. Then $f^{'}(x) = 1 + A\omega cos(\omega x + \varphi), x \in R$.

Since $A\omega cos(\omega x + \varphi) \in [-A\omega, A\omega]$. Next $f^{'}(x) \in [1 - A\omega, 1 + A\omega]$.

Let $f^{'}(x) > 0 => 1 + A\omega cos(\omega x + \varphi) > 0 => 1 - A\omega > 0 => A\omega < 1$.

Let $f^{'}(x) < 0 => 1 + A\omega cos(\omega x + \varphi) < 0 => 1 + A\omega < 0 => A\omega > -1$.

As $-1 \leq A\omega \leq 1$, $f$ is a monotone function. In this case, $\forall A\omega \in [-1, 1], \exists f^{-1}$ is $f$

In this case, $\forall A\omega \in [-1, 1]$, $f$ must be invertible, and the inverse function of $f$ is $f^{-1}$.

Thus, $f$ is bijective if and only if $-1 \leq A\omega \leq 1$.

Finally, when $-1 \leq A\omega \leq 1$, then $f : X {\rightarrow} Y = X + Asin(\omega X + \varphi)$ is a homeomorphism.

**2) Proof: if $A\omega {\in} R$, then $f$ is a local homeomorphism**

as we konw, $h(x) = Asin(\omega x + \varphi), x \in R$ is a periodic function. where $T = \frac{2k\pi}{\omega}$.

Let $2k\pi - \frac{pi}{2} \leq \omega x + \varphi \leq 2k\pi + \frac{pi}{2}, k \in Z$, then $\frac{2k\pi - \frac{pi}{2} - \varphi}{\omega} \leq x \leq \frac{2k\pi + \frac{pi}{2} - \varphi}{\omega}$, now $f$ is strictly increasing.

Let $2k\pi + \frac{pi}{2} \leq \omega x + \varphi \leq 2k\pi + \frac{3pi}{2}, k \in Z$, then $\frac{2k\pi + \frac{pi}{2} - \varphi}{\omega} \leq x \leq \frac{2k\pi + \frac{3pi}{2} - \varphi}{\omega}$, now $f$ is strictly decreasing.

Thus, when $A\omega {\in} R, \forall u \in R, \exists U$ such that $f_U$ is monotone and $f$ is a local homeomorphism.

# D ANALYSIS

## D.1 DETAILED ANALYSIS OF HOW OUR METHOD AFFECTS PART BOUNDARIES

As **Sard's Theorem** stated, when the mapping is smooth and under the condition of general position, the set of points with zero Jacobian determinant will have the property of measure zero, but in practice, the frequently changing sign of the determinant may still lead to the folding phenomenon. Below we carefully analyze how our method affects part boundaries.

Given the mapping $P' = P + A sin(\omega P + \phi)$, we can see it as a transformation from $P$ to $P'$, where $P = (x, y, z)$ and $P' = (x', y', z')$ denote points in 3D space.

**Calculate the Jacobian.**

Set:

$$g(P) = A sin(\omega P + \phi). \tag{6}$$

So the transformation can be written as follows:

$$P' = P + g(P). \tag{7}$$

To calculate the Jacobian $J_{P'}(P)$ of this transformation, we need to take the derivatives with respect to the components of $P'$ with respect to $P$ separately.

1 Write the component forms of $P'$ Suppose each component form is as follows:

$$x' = x + A_x sin(\omega_x x + \phi_x). \tag{8}$$

$$y' = y + A_y sin(\omega_y y + \phi_y). \tag{9}$$

$$z' = z + A_z sin(\omega_z z + \phi_z). \tag{10}$$

Thus, $P' = (x', y', z')$ is obtained by adding $P = (x, y, z)$ to the sine transform of the components.

2 Calculate the elements of the Jacobian matrix.

The Jacobian matrix $J_{P'}(P)$ is of the form:

$$J_{P'}(P) = \begin{bmatrix} \frac{\partial x'}{\partial x} & \frac{\partial x'}{\partial y} & \frac{\partial x'}{\partial z} \\ \frac{\partial y'}{\partial x} & \frac{\partial y'}{\partial y} & \frac{\partial y'}{\partial z} \\ \frac{\partial z'}{\partial x} & \frac{\partial z'}{\partial y} & \frac{\partial z'}{\partial z} \end{bmatrix} \tag{11}$$

Since $x'$ depends only on $x$, $y'$ depends only on $y$, and $z'$ depends only on $z$, the partial derivative matrix is a diagonal matrix.

Diagonal elements are computed:

1) For $\frac{\partial x'}{\partial x}$:

$$\frac{\partial x'}{\partial x} = 1 + A_x \omega_x cos(\omega_x x + \phi_x) \tag{12}$$

2) For $\frac{\partial y'}{\partial y}$:

$$\frac{\partial y'}{\partial y} = 1 + A_y \omega_y cos(\omega_y y + \phi_y) \tag{13}$$

3) For $\frac{\partial z'}{\partial z}$:

$$\frac{\partial x'}{\partial z} = 1 + A_z \omega_z cos(\omega_z z + \phi_z) \tag{14}$$

So the Jacobian matrix is as follows:

$$J_{P'}(P) = \begin{bmatrix} 1 + A_x \omega_x cos(\omega_x x + \phi_x) & 0 & 0 \\ 0 & 1 + A_y \omega_y cos(\omega_y y + \phi_y) & 0 \\ 0 & 0 & 1 + A_z \omega_z cos(\omega_z z + \phi_z) \end{bmatrix} \tag{15}$$

The determinant of this Jacobian is:

$$det(J_{P'}(P)) = (1 + A_x\omega_x cos(\omega_x x + \phi_x)) \cdot (1 + A_y\omega_y cos(\omega_y y + \phi_y)) \cdot (1 + A_z\omega_z cos(\omega_z z + \phi_z)). \quad (16)$$

According to **Sard's Theorem**:

If $det(J_{P'}(P)) > 0$, the map is orientation-preserving near that point, that is, no direction flip occurs locally.

If $det(J_{P'}(P)) < 0$, the map is orientation-reversing near that point, that is, the local direction has been flipped.

Thus:

When $|Aw| < 1$, $det(J_{P'}(P)) > 0$, not affect the part boundaries.

When $|Aw| > 1$, that is, $|A|$ and $|w|$ are large and in the same direction, the determinant may change sign in different regions, which means that the mapping may alternately hold or flip directions in different regions. At this time, the determinant of some regions approaches zero or becomes negative, which may cause the points in local regions to be compressed or collapsed, affecting the part boundaries. In addition, the degree of specific influence on the boundary is also affected by the parameter. When the parameter value does not change much, this influence can be ignored.

As shown in Figure 6, selecting too large a parameter can result in folding, which may affect **part boundaries**. Therefore, in order to ensure topological consistency and no drastic folding occurs, we choose an appropriate $A$ and $w$. As in the ablation experiments in Tables 9 and 10, we finally choose $w = 2.5$ and $A = 0.6$. The model achieves better performance at this time, which means that slight folding is beneficial to the model. However, the above situation does not affect **label consistency**.

## D.2 DETAILED ANALYSIS OF LABEL CONSISTENCY

1) One-to-one mapping between point clouds and labels.

Given point cloud $P = (p_1, p_2, ..., p_n)$. Each point $p_i$ has a corresponding label $l_i$. So you can get a one-to-one correspondence:

$$p_1 \to l_1, p_2 \to l_2, ..., p_n \to l_n. \quad (17)$$

This means that for each point $p_i$, it has a corresponding label $l_i$, which is associated with the geometric position of the point.

2) Definition of homeomorphic mapping.

A homeomorphic map $f(x)$ is a map that preserves topology and changes the positions of points but not the relative relations between them.

If we deform the point cloud by the homeomorphic mapping $f(P)$, then the new point cloud $P'$ is:

$$P' = P + f(P). \quad (18)$$

Therefore, each point position of the new point cloud $P'$ is $p_i' = p_i + f(p_i)$, that is, each point $p_i$ moves to the new position $p_i'$ by mapping $f(x)$.

3) Label consistency.

Since the homeomorphic mapping one-to-one correspondence of points in the point cloud, the index and label of each point are kept consistent after the point cloud is deformed. In other words, the label $l_i$ corresponding to the deformed position $p_i'$ does not change. We can obtain the following relationship:

$$p_1' \to p_1 \to l_1, p_2' \to p_2 \to l_2, ..., p_n' \to p_n \to l_n. \quad (19)$$

Thus:

$$p_1' \to l_1, p_2' \to l_2, ..., p_n' \to l_n. \quad (20)$$

This means that in the deformed point cloud $P'$, the label of each point $p_i'$ remains the same as the label of $p_i$ in the original point cloud $P$.

# E ADDITIONAL EXPERIMENTS

## E.1 MEAN AND STANDARD DEVIATION

Performance oscillation is an essential issue in point cloud benchmarks. However, for a fair comparison with the numbers reported in PointMixup Chen et al. (2020) RSMix Lee et al. (2021), and SageMix Lee et al. (2022), we followed the prevalent evaluation metric in point clouds, which reports the best validation accuracy. Apart from this, to make the experiment fair, like SageMix, we provide the additional results with five runs on OBJ_ONLY Uy et al. (2019) and report the mean and variance of our method, and the experimental results are shown in Table 8.

Table 8: Mean and standard deviation measures on OBJ_ONLY

| Method | Model | | |
| | PointNet | PointNet++ | DGCNN |
| --- | --- | --- | --- |
| Base | 78.56±0.51 | 86.14±0.39 | 85.72±0.44 |
| +PointMixup | 78.88±0.28 | 87.50±0.26 | 86.26±0.34 |
| +RSMix | 77.60±0.56 | 87.30±0.65 | 85.88±0.59 |
| +SageMix | 79.14±0.30 | 88.42±0.26 | 87.32±0.53 |
| **+SinPoint(Ours)** | **82.21±0.36** | **89.83±0.35** | **88.64±0.55** |

## E.2 ABLATION STUDIES AND ANALYSES

**Ablation study of angular velocity** $\omega$**.** We also share the quantitative analysis of the $\omega$ with DGCNN and OBJ_ONLY in Table 9. We observed that SinPoint, with a wide range of $\omega$ (2.5 to 4.5), consistently outperforms PointWOLF 88.8%. At the same time, all cases exceeded the base 86.2%. This shows that the network can learn more representative features after deformation.

Table 9: Ablation study of angular velocity $\omega$.

| $w$ | 0.5 | 1.0 | 1.5 | 2.0 | **2.5** | 3.0 | 3.5 | 4.0 | 4.5 | 5.0 | 7.5 | 10.0 |
| --- | --- | --- | --- | --- | --- | --- | --- | --- | --- | --- | --- | --- |
| Rnad | 88.468 | 88.812 | 88.640 | 88.985 | **90.189** | 89.329 | 89.329 | 88.985 | 88.985 | 88.640 | 89.296 | 88.124 |
| Fixed | 87.673 | 86.403 | 86.747 | 85.714 | 85.714 | 86.747 | 86.403 | 86.231 | 86.919 | 87.091 | 86.059 | 86.059 |

**Ablation study of amplitude** $A$**.** The amplitude $A$ of the Sine function controls the degree of deformation in SinPoint. As shown in Table 10, the larger the $A$, the larger the deformation. However, as shown in Figure 6, too large deformation will lead to the loss of local geometric information. Therefore, we need proper deformation.

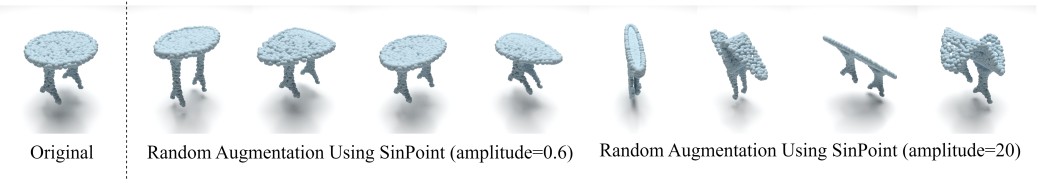

Original     Random Augmentation Using SinPoint (amplitude=0.6)     Random Augmentation Using SinPoint (amplitude=20)

Figure 6: Visualization of ablation results for different parameters of SinPoint.

**Ablation study of anchor points** $k$ **in SinPoint-MSF.** As can be seen from Table 11, when anchor points $k = 4$, the performance reaches 89.845%, but in SinPoint-SSF, the classification performance can reach 90.189%. Although the SinPoint-MSF is effective, exceeding the baseline by 3.6%, the SinPoint-SSF is the best in the classification task. We further verify the performance difference between SinPoint-SSF and SinPoint-MSF in the classification experiment, as shown in Table 12, and we find that SSF performs better.

**Ablation study of amplitude (A) and angular velocity** $\omega$ **sampling.** We explore the effectiveness of amplitude $A$ and angular velocity $\omega$ sampling. Table 13 shows the results with various sampling

Table 10: Ablation study of amplitude $A$.

| $A$ | 0.2 | 0.4 | 0.6 | 0.8 | 1.0 |
|---|---|---|---|---|---|
| OA (%) | 88.985 | 88.812 | **90.189** | 89.329 | 88.985 |
| mAcc (%) | 87.811 | 87.663 | **89.045** | 88.642 | 88.303 |

Table 11: Ablation study of anchor points $k$ in SinPoint-MSF.

| k | 1 | 2 | 3 | 4 | 5 | 6 |
|---|---|---|---|---|---|---|
| RPS | 88.985 | 89.535 | 89.315 | **89.845** | 88.468 | 89.329 |
| FPS | 88.812 | 88.985 | 88.812 | **89.329** | 88.315 | 88.985 |

methods for amplitude $A$ and angular velocity $\omega$. Uniform and Gaussian sampling introduce +3.9% and 2.8% gains over base DGCNN. The OA with Uniform sampling is 2.1% higher than Gaussian sampling, which means that uniform sampling leads to greater diversity and maximizes model performance.

**3D part segmentation performance under Various Baselines.** The effectiveness of SinPoint is further validated across a variety of network architectures in ShapeNetPart Yi et al. (2016), including PointNet Qi et al. (2017a), PointNet++ Qi et al. (2017b), DGCNN Wang et al. (2019b), CurveNet Xiang et al. (2021), 3DGCN Lin et al. (2021), GDANet Xu et al. (2021c), PointMLP Ma et al. (2022), SPoTr Park et al. (2023), PointMetaBase Lin et al. (2023) and DeLA Chen et al. (2023). Table 14 shows that SinPoint has a consistent improvement of mean Inter-over-Union (mIoU) over the baselines (+0.1∼1.0%).

**Performance on scene segmentation.** We added additional experiments to the S3DIS Armeni et al. (2016) and SemanticKITTI Behley et al. (2019) datasets. As shown in Table 15, our SinPoint is still able to improve MinkNet Choy et al. (2019) performance.

E.3 VISUALIZATION

**Convergence analysis.** As shown in Figure 7, our SinPoint demonstrates faster convergence during the training phase and achieves higher accuracy than the baseline. Consistent performance improvement is achieved under various parameter Settings.

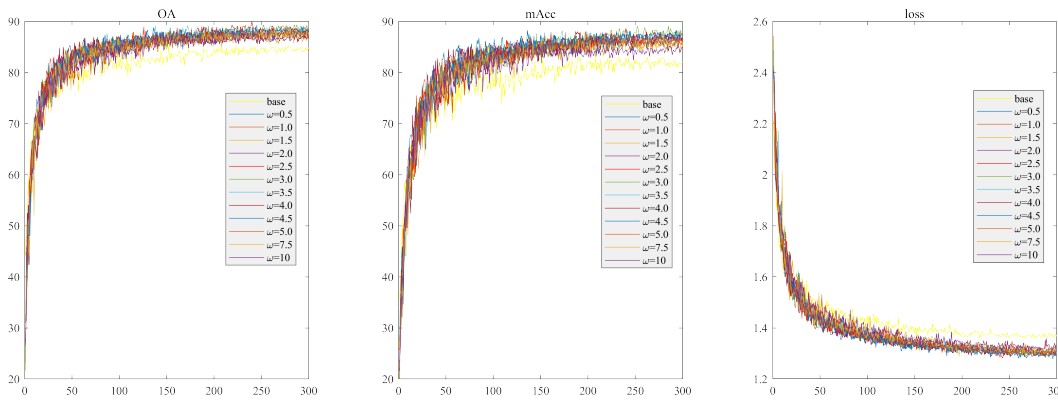

Figure 7: Convergence curve during the model training phase. Our SinPoint has a faster convergence speed and higher convergence accuracy than the baseline.

**Training efficiency.** As shown in Table 16, Our SinPoint achieves the best performance while reducing the time per training epoch. Notably, the training time is reduced by 10 times compared to PointAugment, 6 times compared to SageMix, and 2 times compared to PointWOLF.

Table 12: Ablation study of SinPoint-SSF and SinPoint-MSF on ScanObjectNN.

| Method | SinPoint-SSF | SinPoint-MSF |
|---|---|---|
| OA | 89.845 | **90.189** |

Table 13: Ablation study on amplitude $A$ and angular velocity $\omega$ sampling.

| | OA | mAcc |
|---|---|---|
| base | 85.829±0.296 | 83.375±0.395 |
| **Uniform** | **89.759±0.431** | **88.636±0.445** |
| Gaussian | 87.607±0.344 | 85.494±0.409 |

**Qualitative results of SinPoint.** In Figure 8, we give a visualization of more augmented samples. In Figure 9, we present an augmented sample visualization comparison between SinPoint-SSF and SinPoint-MSF.

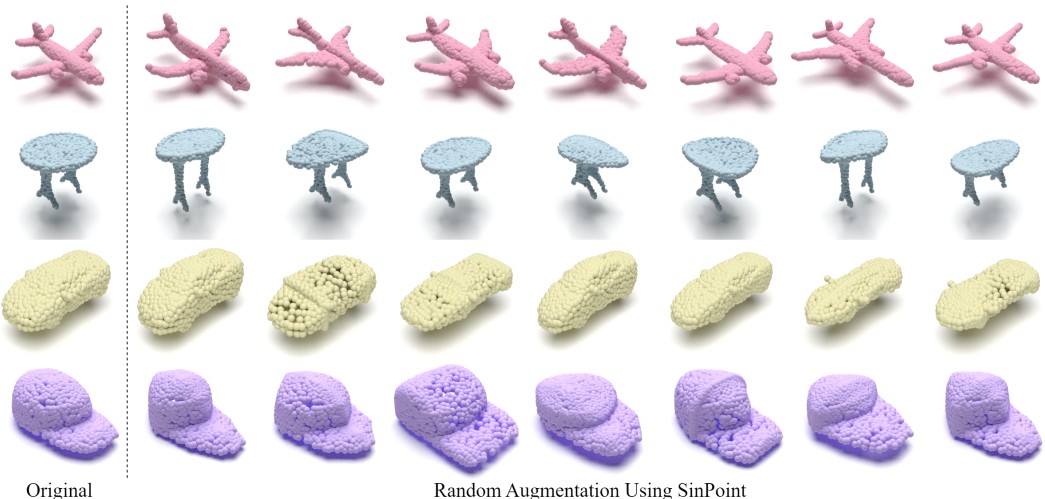

Original                                    Random Augmentation Using SinPoint

Figure 8: Augmented point clouds using SinPoint. In each row, the left-most sample is the original, and the remaining samples are its transformed results.

**Qualitative results compare SinPoint with PointWOLF.** We compare our SinPoint with Point-WOLF in geometric diversity and topological consistency of point clouds. As can be seen from Figure 10, our SinPoint is entirely superior to PointWOLF and does not require **AugTune** Kim et al. (2021). The results generated by our SinPoint are more in line with the real world. On the contrary, many of the results generated by PointWOLF are out of the reality.

### E.4    DISCUSSION AND FUTURE WORK

In the future, we will apply SinPoint to more tasks, such as feature space augmentation, few-shot learning Liu et al. (2019); Qi et al. (2017a), semantic segmentation Chen et al. (2019b); Xu et al. (2021a); Wang et al. (2019a), object detection Taha et al. (2020); Zhao et al. (2021); Sugimura et al. (2020), etc. It is worth noting, however, that different tasks require different considerations. For example, in few-shot learning, SinPoint maximizes the diversity of training data when samples are extremely scarce. For object detection, SinPoint can generate richer 3D transformations for various object instances in a 3D scene, and so on. Therefore, SinPoint will be easily extended to other tasks.

Table 14: Part segmentation performance in various architectures on ShapeNetPart Yi et al. (2016). The proposed SinPoint shows consistent improvements over baselines.

| Model | PointNet | PointNet++ | DGCNN | CurveNet | 3DGCN | GDANet | PointMLP | SPoTr | PointMetaBase | DeLA |
|---|---|---|---|---|---|---|---|---|---|---|
| Base | 83.5 | 84.8 | 84.8 | 86.6 | 86.4 | 86.1 | 85.8 | 87.0 | 86.9 | 87.0 |
| +SinPoint | 84.4 (↑ 0.9) | 85.8 (↑ 1.0) | 85.5 (↑ 0.7) | 86.8 (↑ 0.2) | 86.6 (↑ 0.2) | 86.2 (↑ 0.1) | 86.1 (↑ 0.3) | 87.2 (↑ 0.2) | 87.3 (↑ 0.4) | 87.4 (↑ 0.4) |

Table 15: SinPoint on S3DIS and SemanticKITTI.

| Method | S3DIS (mIoU) | SemanticKITTI (mIoU) |
|---|---|---|
| MinkNet | 64.8 | 55.9 |
| **+SinPoint (Ours)** | **65.4(+0.6)** | **63.5(+7.6)** |

# F    COMPARE TO OTHER METHODS

## F.1    CONVENTIONAL DATA AUGMENTATION

A Conventional Data Augmentation (CDA) Qi et al. (2017a;b); Wang et al. (2019b) for point clouds applies a global similarity transformation (e.g., scaling, rotation, and translation) and point-wise jittering. Given a set of points $P = \{p_i | i = 1, 2, ..., N\}$, where $N$ represents the number of points in the Euclidean space $(x, y, z)$. The augmented point cloud $P^{'}$ is given as follows:

$$P^{'} = SRP + B. \tag{21}$$

where $S > 0$ is a scaling factor, $R$ is a 3D rotation matrix, and $B \in R^{N \times 3}$ is a translation matrix with global translation and point-wise jittering. Typically, $R$ is an extrinsic rotation parameterized by a uniformly drawn Euler angle for the up-axis orientation. Scaling and translation factors are uniformly drawn from an interval, and point-wise jittering vectors are sampled from a truncated Gaussian distribution.

Obviously, when $B$ does not exist, CDA is a rigid transformation, and when $B$ exists, CDA is simply a similarity transformation with jitter. Thus, CDA cannot simulate diverse shapes and deformable objects, and the enhanced sample has poor diversity.

## F.2    MIX-AUGMENTATION

Several works Chen et al. (2020); Lee et al. (2021) tried to leverage the Mixup in point cloud. PointMixup linearly interpolates two point clouds by

$$P^{'} = \{\lambda p_i^{\alpha} + (1 - \lambda) p_{\phi^*(i)}^{\beta}\}_i^n, y^{'} = \lambda y^{\alpha} + (1 - \lambda) y^{\beta}. \tag{22}$$

$$\phi^* = \arg\min_{\phi \in \Phi} \sum_{i=1}^{n} \|p_i^{\alpha} + p_{\phi(i)}^{\beta}\|_2 \tag{23}$$

where $P^t = \{p_1^t, ..., p_n^t\}$ is the set of points with $t \in \{\alpha, \beta\}$, $n$ is the number of points, and $\phi^* : \{1, ..., n\} \rightarrow \{1, ..., n\}$ is the optimal bijective assignment between two point clouds. In RSMix Lee et al. (2021), they generate an augmented sample by merging the subsets of two objects, defined as $P^{'} = (P^{\alpha} - S^{\alpha}) \cup S^{\beta \rightarrow \alpha}$, where $S^t \subset P^t$ $t$ is the rigid subset and $S^{\beta \rightarrow \alpha}$ denotes $S^{\beta}$ translated to the center of $S^{\alpha}$. SageMix sequentially selects the query point based on the saliency scores to improve the above method.

Table 16: Comparisons of the training efficiency on ModelNet40 using PointNet.

| Method | PointAugment | SageMix | PointWOLF | SinPoint (Ours) |
|---|---|---|---|---|
| OA (%) | 74.4 | 79.5 | 78.7 | **82.6** |
| Time (sec) | 84 | 51 | 15 | **8** |

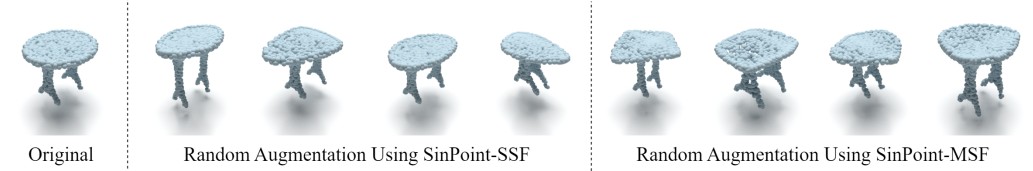

Figure 9: Visualization comparison of our SinPoint. The left is SinPoint-SSF, right is SinPoint-MSF. SinPoint-MSF can produce varying degrees of local deformation in different regions.

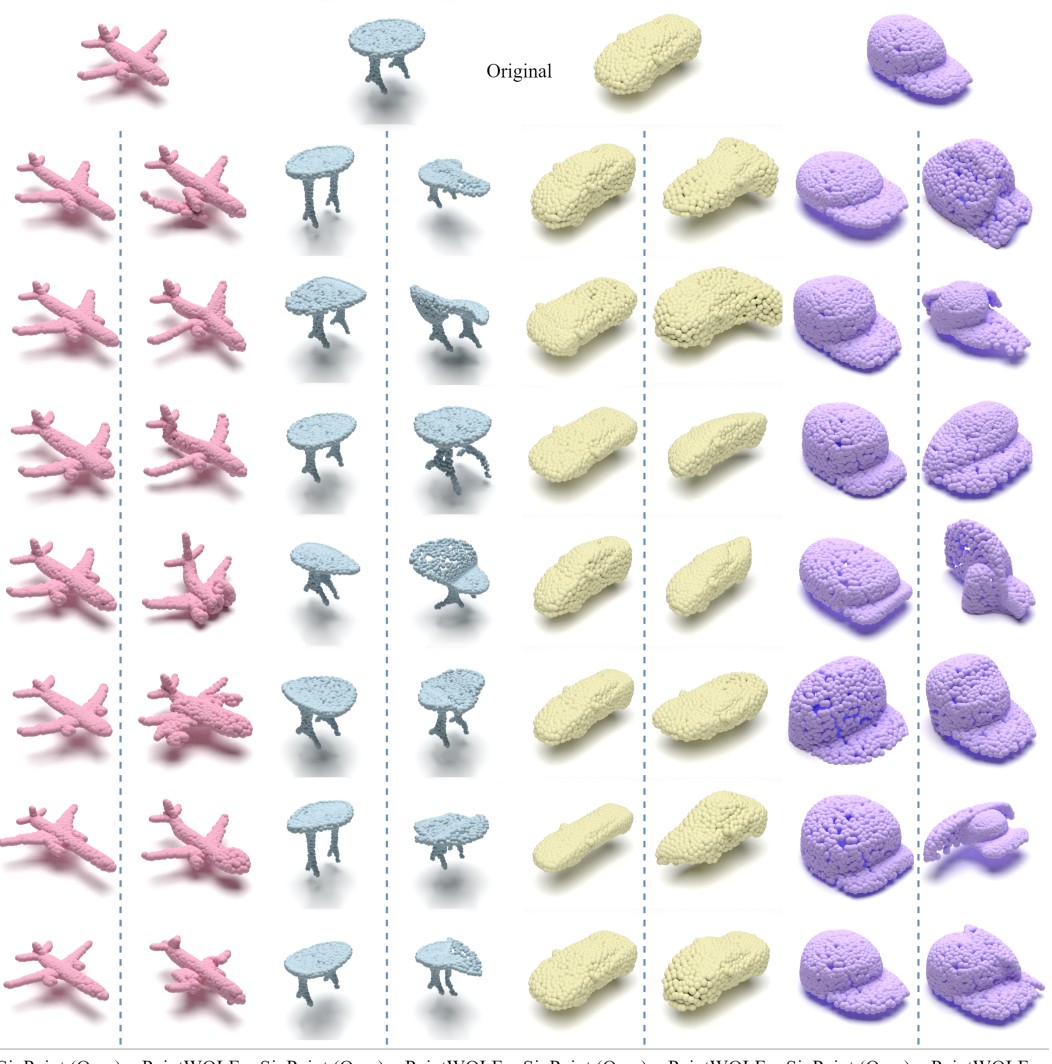

Figure 10: Visualization comparison of our SinPoint with PointWOLF. The top original point cloud, each shape left is SinPoint, right is PointWOLF. The samples generated by our SinPoint are more diverse and realistic.

Although these methods have shown that Mixup is effective for point clouds, some limitations have remained unresolved: loss of original structures, discontinuity at the boundary, and loss of discriminative regions. Therefore, although the method based on mixup can increase the diversity of samples by mixup different samples, it destroys the point cloud structure.

### F.3 Self-Augmentation

A representative method is PointWOLF. PointWOLF generates deformation for point clouds by a convex combination of multiple transformations with smoothly varying weights. PointWOLF first selects several anchor points and locates random local transformations (e.g., similarity transformations) at the anchor points. Based on the distance from a point in the input to the anchor points, PointWOLF differentially applies the local transformations. The smoothly varying weights based on the distance to the anchor points allow spatially continuous augmentation and generate realistic samples. Given an anchor point $p_j^A \in P^A$. the local transformation for an input point $p_j^A \in P_i$ can be written as:

$$p_i^j = S_j R_j (p_i - p_j^A) + B_j + p_j^A. \tag{24}$$

where $R_j$ , $S_j$ and $B_j$ are rotation matrix, scaling matrix and translation vector $bj$ respectively which specifically correspond to $p_j^A$. $S$ is a diagonal matrix with three positive real values, i.e., $S = diag(s_x, s_y, s_z)$ to allow different scaling factors for different axes.

Due to the local rotation and translation, the local separation from the main body will cause the topological structure of the point cloud to change, so PointWOLF is not a homeomorphism and cannot guarantee the topological consistency.

Meanwhile, AugTune is also required due to the poor performance of the PointWOLF direct transform. For $N$ points and $M$ anchor points, the time complexity of PointWOLF is $O(MN) + O(N)$. However, our SinPoint can produce realistic augmented data without interpolation, and our SinPoint time complexity is only $O(N)/O(MN)$, which can reduce the amount of computation.

