# OpenReview forum: "SinPoint: A Novel Topological Consistent Augmentation for Point Cloud Understanding"
_ICLR.cc/2025/Conference — ICLR 2025 Conference Withdrawn Submission_

### Official Review · Reviewer_2azz · 2024-10-24

**Soundness:** 3
**Presentation:** 3
**Contribution:** 3
**Rating:** 8
**Confidence:** 3

**Summary:**

This paper presents a novel data augmentation method called SinPoint, which enhances the geometric diversity of point clouds while preserving the topological structure of the original point cloud through a homeomorphism. Experiments conducted on multiple datasets and models demonstrate the effectiveness of this approach.

**Strengths:**

The SinPoint method proposed in this paper effectively enhances performance across multiple datasets and models, and it is both concise and efficient. Additionally, the article is well-structured, clearly articulated, and includes rigorous mathematical proofs. The experiments are comprehensive, and supported by sufficient visualization results.

**Weaknesses:**

1. There are some minor errors in the details. In Table 1, the results for SinPoint on PointNet++ and ModelNet40 are calculated incorrectly. Additionally, similar errors are present in Table 2 for PCT and PointMLP. I suggest that the authors carefully review the results in these tables.

**Questions:**

1. It is recommended that the authors thoroughly review the results in the tables and make corrections in subsequent versions.

2. Figure 3 indicates that random angular velocities may lead to uncontrolled variations. Are there detailed experimental results demonstrating the performance differences between random angular velocities and fixed angular velocities? For instance, it would be helpful to report the accuracy on ModelNet40.

3. The results in Table 5 suggest that SinPoint can be combined with other data augmentation methods. Would it be possible to integrate more advanced mixup techniques to further enhance performance?

---

> ### Author Response · Authors · 2024-11-19
> **Author Response to Reviewer 2azz**
>
> **Thanks for your time and insightful reviews. We responded in detail as follows:**
>
> **Q1:** There are some minor errors in the details. In Table 1, the results for SinPoint on PointNet++ and ModelNet40 are calculated incorrectly. Additionally, similar errors are present in Table 2 for PCT and PointMLP. I suggest that the authors carefully review the results in these tables. It is recommended that the authors thoroughly review the results in the tables and make corrections in subsequent versions.
>
> **A1:** Thanks for your careful review, we have reviewed all the forms and fixed the problem. **We have uploaded a revision of our paper and marked it in blue.**
>
> **Q2:** Figure 3 indicates that random angular velocities may lead to uncontrolled variations. Are there detailed experimental results demonstrating the performance differences between random angular velocities and fixed angular velocities? For instance, it would be helpful to report the accuracy on ModelNet40.
>
> **A2:** We give **a comparison of the different angular velocities in Table R1.** As you can see, different angular velocities cause oscillations in performance. When a fixed angular velocity is chosen, the diversity of deformation is lost. The performance of the model is also lost. **We have uploaded a revision of our paper and marked it in blue.**
>
> **Table R1: Random angular velocities and fixed angular velocities.**
> | Method | 0.5 | 1.0 | 1.5 | 2.0 | 2.5 | 3.0 | 3.5 | 4.0 | 4.5 | 5.0 | 7.5 | 10.0 |
> |:-|:-:|:-:|:-:|:-:|:-:|:-:|:-:|:-:|:-:|:-:|:-:|:-:|
> | **Rand w** | 88.468 | 88.812 | 88.640 | 88.985 | 90.189 | 89.329 | 89.329 | 88.985 | 88.985 | 88.640 | 89.296 | 88.124 |
> | Fixed w | 87.673 | 86.403 | 86.747 | 85.714 | 85.714 | 86.747 | 86.403 | 86.231 | 86.919 | 87.091 | 86.059 | 86.059 |
>
> **Q3:** The results in Table 5 suggest that SinPoint can be combined with other data augmentation methods. Would it be possible to integrate more advanced mixup techniques to further enhance performance?
>
> **A3:** Yes, it is possible to integrate our SinPoint with more advanced mixup techniques. **However, it may not help to further enhance performance, for example: WolfMix (PMLR 2022) is a successful example of data augmentation by combining PointWOLF with RSMix.** We have compared this hybrid approach in Table 1. However, this makes the method more complex and performs only better than RSMix.
>
> **According to your suggestion, we conducted a preliminary experiment by combining the SageMix with our SinPoint. As shown in Table R2, a 1.2 improvement in accuracy compared to SageMix. However, performance slightly decreased when compared to SinPoint.**
>
> We believe that the strategy of combining these methods may need further improved to achieve better performance. **We sincerely appreciate the reviewers' insightful questions, and we plan to continue exploring and advancing research on this hybrid approach in future work.**
>
> **Table R2: A hybrid method of SageMix and SinPoint (Ours) on OBJ_ONLY.**
> | DGCNN | OA (%) |
> |:---|:---:|
> | +SageMix | 88.0 |
> | **+SinPoint (Ours)** | **90.2** |
> | **+SinPoint+SageMix** | **89.2** |

---

> > ### Comment · Reviewer_2azz · 2024-11-24
> >
> > Thank you for your reply and I will keep my score

---

> > > ### Author Response · Authors · 2024-11-25
> > > **Thank You Vey Much for Your Response**
> > >
> > > Dear Reviewer 2azz,
> > >
> > > Thank you very much for the follow-up!
> > >
> > > We appreciate the discussion, your feedback, and your suggestions. Many thanks for your time and effort.
> > >
> > > Best and sincere wishes,
> > >
> > > The authors

---

### Official Review · Reviewer_pDnW · 2024-11-03

**Soundness:** 2
**Presentation:** 3
**Contribution:** 2
**Rating:** 6
**Confidence:** 4

**Summary:**

The paper introduces a data augmentation method for point cloud understanding, which utilize Sine function to generate smooth displacements and preserve the topological structure of the original points via a homeomorphism. The authors evaluate the proposed method with different backbones on object classification and part segmentation tasks to demonstrate the effectiveness.

**Strengths:**

+ The authors propose a simple but effective data augmentation method in some scenarios .
+ The writing and presentation is clear.

**Weaknesses:**

- While homeomorphism can theoretically provide a wide range of continuous transformations for data augmentation, in practice, they can distort essential geometric features (can preserve topological properties but not geometric ones like distances, angles, and curvature), introduce unrealistic data (can warp objects into unnatural shapes that do not exist in the real world as shown in Figure 2, 6, 8 and 9), and complicate the learning process for models in part segmentation (can alter these relationships by stretching or compressing certain regions, leading to confusion in part boundaries and potential mislabeling; then providing incorrect training signals and degrading the model's performance as shown in Table 6 and 7).

- Limited Experimental Evaluation:
  + For the synthetic dataset in the classification task, the proposed method is effective when combined with older methods but less so with recent ones, as shown in Table 2.
  + Insufficient Real-World Testing: The authors only tested on the easiest subset (OBJ_ONLY) of the ScanObjectNN dataset (Tables 3 and 4), which does not confirm the method's superior performance with complex and noisy data (PB_T50_RS of ScanObjectNN or other real-world datasets).
  + In the part segmentation task, the proposed method performs on par with other methods. Since segmentation requires point-wise labels rather than whole shape labels, the effectiveness of homeomorphism-based augmentation is limited in this context (as mentioned above).

**Questions:**

- For the part segmentation, could the authors provide a more detailed analysis of how their method affects part boundaries and label consistency?
- The authors should evaluate their method on more challenging datasets, such as classification on PB_T50_RS of ScanObjectNN, semantic segmentation tasks in S3DIS (real-world, room-scale environments) and SemanticKITTI (outdoor, dynamic, real-world scenarios), to demonstrate its robustness and applicability to complex data.
- The authors should provide the extra model size, parameters, throughput when combining the proposed method with other backbones.

---

> ### Author Response · Authors · 2024-11-19
> **Author Response to Reviewer pDnW (part 1/3)**
>
> **Thanks for your time and insightful reviews. We responded in detail as follows:**
>
> **Q1:** While homeomorphism can theoretically provide a wide range of continuous transformations for data augmentation, in practice, they can distort essential geometric features (can preserve topological properties but not geometric ones like distances, angles, and curvature), introduce unrealistic data (can warp objects into unnatural shapes that do not exist in the real world as shown in Figure 2, 6, 8 and 9),……
>
> **A1:** The purpose of our SinPoint is to increase the diversity of geometric structures while maintaining topological consistency (see Abstract). The data may initially appear to warp objects into unnatural shapes, but they're still meaningful for learning. The main reasons are as follows:
>
> 1.**Need for Data Augmentation:** To enhance the model's generalization ability, we introduce various types of deformations and data transformations. These augmentations not only simulate the diversity of inputs encountered in real-world applications but also help the model learn to perform well across a broader range of scenarios. **For example: PointWOLF will initially appear to warp objects into unnatural shapes, as shown in Figure 1.**
>
> 2.**Improvement of Model Robustness:** Although the deformed data may not align with the physical constraints of the real world, it effectively enhances the model's ability to adapt to shape variations. This allows the model to become more robust to unusual or extreme samples.
>
> 3.**Validation of Experimental Results:** As demonstrated in Tables 1, 2, 3, 6, 7, and 14, the experimental results clearly show significant improvements in model performance after incorporating these augmented data.
>
> 4.**Rationality of the Data:** Our intention is not to replicate every detail of the real world but to design deformations that simulate extreme or uncommon scenarios. These situations are useful for improving the model's ability to generalize and predict when faced with real-world data. We believe this approach is aligned with the data augmentation strategies commonly employed in the literature.
>
> **In summary, while these augmented data may not perfectly resemble the typical morphology of real-world objects, they play a crucial role in model training and significantly enhance performance.**
>
> **Q2:** ……and complicate the learning process for models in part segmentation (can alter these relationships by stretching or compressing certain regions, leading to confusion in part boundaries and potential mislabeling; then providing incorrect training signals and degrading the model's performance as shown in Table 6 and 7).
>
> **A2:** Thanks for your interest in our research and for your valuable comments regarding the potential issues in the data augmentation process.
>
> **To address these issues, we have adjusted parameters of our SinPoint to prevent excessive geometric transformations of the point cloud.** Specifically, we introduce **more conservative parameters** to ensure that the boundaries of the transformed parts remain as accurate as possible, **avoiding extreme distortion**. **As illustrated in Figure 6, we observe that overly large deformation parameters can lead to confusion in part boundaries and potential mislabeling.** Thus, we specifically avoid such extreme deformations.
>
> **Furthermore**, our experimental results, **as shown in Tables 8 and 9**, indicate that moderate augmentation—**using parameters A = 0.6 and w = 2.5**—provides the best balance for improving model robustness.
>
> **Q3:** For the synthetic dataset in the classification task, the proposed method is effective when combined with older methods but less so with recent ones, as shown in Table 2.
>
> **A3:** This situation **is quite common in machine learning**. Since there is considerable potential for improvement in the older methods, **greater performance gains can be achieved.** The performance of the recent methods is approaching saturation and is nearing the theoretical upper bound of the ModelNet40 dataset. **Thus, we combined our method with some classical methods.**
>
> **Q4:** Insufficient Real-World Testing: The authors only tested on the easiest subset (OBJ_ONLY) of the ScanObjectNN dataset (Tables 3 and 4), which does not confirm the method's superior performance with complex and noisy data (PB_T50_RS of ScanObjectNN or other real-world datasets).
>
> **A4: We have shown the result of our SinPoint on the PB_T50_RS dataset in Table 1. I am sorry that the title of in Table 3 is mislabeled. The correct dataset name is PB_T50_RS. We have uploaded a revision of our paper and marked it in green.**

---

> ### Author Response · Authors · 2024-11-19
> **Author Response to Reviewer pDnW (part 2/3)**
>
> **Q5:** In the part segmentation task, the proposed method performs on par with other methods. Since segmentation requires point-wise labels rather than whole shape labels, the effectiveness of homeomorphism-based augmentation is limited in this context (as mentioned above).
>
> **A5:** As **shown in Table 6, 7 and 14, comparing with baseline, all existing methods show limited improvement in part segmentation task**. However, **our method still achieved the highest performance improvement**. So **homeomorphism-based augmentation is our advantage, not our limitation**. This is because our deformation maintains topological consistency and increases the diversity of shapes, **enabling the model to learn more discriminative features.**
>
> **Q6:** For the part segmentation, could the authors provide a more detailed analysis of how their method affects part boundaries?
>
> **A6: Detailed analysis of how our method affects part boundaries.**
>
> As **Sard's Theorem** stated, when the mapping is smooth and under the condition of general position, the set of points with zero Jacobian determinant will have the property of measure zero, but in practice, the frequently changing sign of the determinant may still lead to the folding phenomenon. Below we carefully analyze how our method affects part boundaries..
>
> Given the mapping $P' = P+ Asin(\omega P + \phi)$, we can see it as a transformation from $P$ to $P'$, where $P =(x,y,z)$ and $P' =(x',y',z')$ denote points in 3D space.
>
> **Calculate the Jacobian**
>
> Set $g(P) = Asin(\omega P + \phi)$. So the transformation can be written as follows: $P' = P+ g(P)$.
>
> To calculate the Jacobian $J_{P'}(P)$ of this transformation, we need to take the derivatives with respect to the components of $P'$ with respect to $P$ separately.
>
> **1) Write the component forms of $P'$ Suppose each component form is as follows:**
>
> $x' = x + A_{x}sin(\omega_{x}x + \phi_{x}), y' = y + A_{y}sin(\omega_{y}y + \phi_{y}), z' = z + A_{z}sin(\omega_{z}z + \phi_{z}).$
>
> Thus, $P'=(x',y',z')$ is obtained by adding $P=(x,y,z)$ to the sine transform of the components.
>
> **2) Calculate the elements of the Jacobian matrix.**
>
> **The Jacobian matrix $J_{P'}(P)$ is of the form:**
>
> $J_{P'}(P) = [\frac{\partial_{x'}}{\partial_{x}}, \frac{\partial_{x'}}{\partial_{y}}, \frac{\partial_{x'}}{\partial_{z}}; \frac{\partial_{y'}}{\partial_{x}}, \frac{\partial_{y'}}{\partial_{y}}, \frac{\partial_{y'}}{\partial_{z}}; \frac{\partial_{z'}}{\partial_{x}}, \frac{\partial_{z'}}{\partial_{y}}, \frac{\partial_{z'}}{\partial_{z}}]$
>
> Since $x'$ depends only on $x$, $y'$ depends only on $y$, and $z'$ depends only on $z$, the partial derivative matrix is a diagonal matrix.
>
> **Diagonal elements are computed:**
>
> $\frac{\partial_{x'}}{\partial_{x}} = 1+A_{x}\omega_{x}cos(\omega_{x}x + \phi_{x})$
>
> $\frac{\partial_{y'}}{\partial_{y}} = 1+A_{y}\omega_{y}cos(\omega_{y}y + \phi_{y})$
>
> $\frac{\partial_{x'}}{\partial_{z}} = 1+A_{z}\omega_{z}cos(\omega_{z}z + \phi_{z})$
>
> **So the Jacobian matrix is as follows:**
>
> $J_{P'}(P) = [1+A_{x}\omega_{x}cos(\omega_{x}x + \phi_{x}), 0, 0; 0, 1+A_{y}\omega_{y}cos(\omega_{y}y + \phi_{y}), 0; 0, 0, 1+A_{z}\omega_{z}cos(\omega_{z}z + \phi_{z})]$
>
> The determinant of this Jacobian is: $det(J_{P'}(P)) = (1+A_{x}\omega_{x}cos(\omega_{x}x + \phi_{x})) \cdot (1+A_{y}\omega_{y}cos(\omega_{y}y + \phi_{y})) \cdot (1+A_{z}\omega_{z}cos(\omega_{z}z + \phi_{z})).$
>
> According to **Sard's Theorem**:
>
> If $det(J_{P'}(P))>0$, the map is orientation-preserving near that point, that is, no direction flip occurs locally.
>
> If $det(J_{P'}(P))<0$, the map is orientation-reversing near that point, that is, the local direction has been flipped.
>
> Thus:
>
> When $|Aw|<1$, $det(J_{P'}(P))>0$, not affect the part boundaries.
>
> When $|Aw|>1$, that is, $|A|$ and $|w|$ are large and in the same direction, the determinant may change sign in different regions, which means that the mapping may alternately hold or flip directions in different regions. At this time, the determinant of some regions approaches zero or becomes negative, which may cause the points in local regions to be compressed or collapsed, affecting the part boundaries. **In addition, the degree of specific influence on the boundary is also affected by the parameter. When the parameter value does not change much, this influence can be ignored.**
>
> **As shown in Figure 6, selecting too large a parameter can result in folding, which may affect part boundaries. **Therefore, in order to ensure topological consistency and no drastic folding occurs, we choose an appropriate $A$ and $w$. As in the ablation experiments in Tables 9 and 10, we finally choose $w=2.5$ and $A=0.6$.** The model achieves better performance at this time, which means that slight folding is beneficial to the model. However, the above situation does not affect label consistency.**

---

> ### Author Response · Authors · 2024-11-19
> **Author Response to Reviewer pDnW (part 3/3)**
>
> **Q7:** For the part segmentation, could the authors provide a more detailed analysis of label consistency?
>
> **A7: Detailed analysis of label consistency.**
>
> **1) One-to-one mapping between point clouds and labels.**
>
> Given point cloud $P = (p_1,p_2,...,p_n)$. Each point $p_i$ has a corresponding label $l_i$. So you can get a one-to-one correspondence: $p_1 \to l_1, p_2 \to l_2,...,p_n \to l_n$. This means that for each point $p_i$, it has a corresponding label $l_i$, which is associated with the geometric position of the point.
>
> **2) Definition of homeomorphic mapping.**
>
> A homeomorphic map $f(x)$ is a map that preserves topology and changes the positions of points but not the relative relations between them. If we deform the point cloud by the homeomorphic mapping $f(P)$, then the new point cloud $P'$ is: $P' = P + f(P)$. Therefore, each point position of the new point cloud $P'$ is $p_{i}'=p_{i} + f(p_{i})$, that is, each point $p_{i}$ moves to the new position $p_{i}'$ by mapping $f(x)$.
>
> **3) Label consistency.**
>
> Since the homeomorphic mapping one-to-one correspondence of points in the point cloud, the index and label of each point are kept consistent after the point cloud is deformed. **In other words, the label $l_i$ corresponding to the deformed position $p_i'$ does not change. We can obtain the following relationship: $p_1' \to p_1 \to l_1, p_2' \to p_2 \to l_2,...,p_n' \to p_n \to l_n$. Thus, $p_1' \to l_1, p_2' \to l_2,...,p_n' \to l_n$. This means that in the deformed point cloud $P'$, the label of each point $p_i'$ remains the same as the label of $p_i$ in the original point cloud $P$.**
>
> **Q8:** The authors should evaluate their method on more challenging datasets, such as classification on PB_T50_RS of ScanObjectNN, semantic segmentation tasks in S3DIS (real-world, room-scale environments) and SemanticKITTI (outdoor, dynamic, real-world scenarios), to demonstrate its robustness and applicability to complex data.
>
> **A8:** In fact, **we have evaluated our SinPoint on more challenging datasets (PB_T50_RS of ScanObjectNN), as shown in Table 1 and 3.** According to your suggestion, **we added additional experiments on the S3DIS and SemanticKITTI dataset. As shown in Table R1, our SinPoint is still able to improve the performance of baseline.**
>
> **Table R1: SinPoint on S3DIS and SemanticKITTI.**
> | Method | S3DIS (mIoU) | SemanticKITTI (mIoU) |
> |:---|:---:|:---:|
> | MinkNet | 64.8 | 55.9 |
> | **+SinPoint (Ours)** | **65.4(+0.6)** | **63.5(+7.6)** |
>
> **Q9:** The authors should provide the extra model size, parameters, throughput when combining the proposed method with other backbones.
>
> **A9:** In our SinPoint, **model size, parameters and throughput are exactly the same as baseline,** because we are a non-parameter data augmentation method, and its aim is to generate augmented samples. **Therefore, our SinPoint do not change model size, and will not bring any parameters. Meanwhile, it also does not affect the throughput of baseline.** If you are interested in model size, parameters and throughput, you can look them up in baselines' papers.

---

> ### Author Response · Authors · 2024-11-25
> **End of Discussion Approaching**
>
> Dear Reviewer pDnW:
>
> We sincerely appreciate your detailed review and valuable feedback. We have carefully addressed your concerns and made revisions accordingly.
>
> Given that the end of the discussion period is approaching, we would like to ask if there are still any areas that seem unclear or if you have additional questions. We are more than happy to offer additional clarifications.
>
> The current work on point cloud augmentation is largely limited to mimicking image augmentation, while our SinPoint represents a significant innovation and achieves state-of-the-art performance. We hope that these advancements are recognized and valued.
>
> We appreciate the discussion, your feedback, and your suggestions. Many thanks for your time and effort.
>
> Best and sincere wishes,
>
> The authors

---

> > ### Comment · Reviewer_pDnW · 2024-11-28
> >
> > Dear authors,
> >
> > Thank you for your detailed response and it took me longer time to read all of them. Even though, there are still some points that weaken this paper, however, I think if the authors include the new content during discussion phase to the paper, then it is okay for acceptance. I will raise my scores.

---

> > > ### Author Response · Authors · 2024-11-28
> > > **Thank You Vey Much for Your Response**
> > >
> > > Dear Reviewer pDnW,
> > >
> > > Thank you very much for the follow-up!
> > >
> > > We appreciate the discussion, your feedback, and your suggestions. Many thanks for your evaluable time and effort.
> > >
> > > Thank you for your positive feedback and for considering an increased score for our paper! We sincerely appreciate your constructive comments and thorough review.
> > >
> > > Thank you again for your recognition of our work. We have uploaded the revised version according to your suggestion.
> > >
> > > Best and sincere wishes,
> > >
> > > The authors

---

### Official Review · Reviewer_MMqo · 2024-11-04

**Soundness:** 3
**Presentation:** 3
**Contribution:** 3
**Rating:** 6
**Confidence:** 5

**Summary:**

The paper proposes a novel point cloud data augmentation technique called SinPoint. It is based on a homomorphism and maintains topological consistency while increasing the geometric diversity of samples (perturbs the local structure using a Sine function). The approach has 1) a Single Sine Function (SinPoint-SSF) to deform the point cloud with origin as the initial phase and 2) a Multiple Sine Function (SinPoint-MSF) with different anchor points as the initial phase. Richer deformations are achieved from the superimposition of the sine transforms of different parameters. The paper demonstrates the effectiveness of the proposed framework by showing consistent improvements over SOTA augmentation methods on both synthetic and real-world datasets in 3D shape classification and part segmentation tasks. The proposed SinPoint improves the generalization and robustness of several classification models and segmentation (PointNet, PointNet++, and DGCNN) models.

**Strengths:**

+ Most recent 3D point cloud augmentation methods focus on classification. The proposed approach also focuses on segmentation.
+ Most existing augmentation methods are proposed as an alternative to conventional data augmentation (CDA). The proposed approach is in addition to CDA and allows the simultaneous use of other augmentation techniques.
+ The paper is easy to read and well-written, with ample references to recent works.
+ The experiments sections provide ample proof of the proposed approach's impact on various SOTA point cloud classification methods.

**Weaknesses:**

- The impact of the proposed augmentation techniques has been demonstrated on a limited number of segmentation backbones (precisely PointNet, PointNet++, and DGCNN). These backbones are almost five-year-old methods and have ample room to show the impact. The proposed work does not show its effects in recent methods (PointNeXt onwards), making it unclear if the proposed augmentation approach is relevant to the current state-of-the-art segmentation methods.
- Figures 8 and 9 in the supplementary pages (page 19) show that the second augmented table generated (using SinPoint SSF) shows that the table top surface has a concave bend so that items placed on it would roll over. Such samples are not realistic or meaningful.
- Inconsistency in using recent models to demonstrate the improvement in classification accuracy for different datasets. For example, PointNeXt, PointMetaBase, and SpoTr are selected to show the improvement in results for the ScanObjectNN dataset's OBJ_ONLY variant in Table 3; the same models are ignored to show the upgrades for the ModelNet40 dataset in Table 2. And vice versa, when considering models in Table 2 that were used to show improvements for the ModelNet40 dataset, they are ignored when showing improvements on ScanObjectNN dataset variants.
- Some of the SOTA models have both classification and segmentation backbones, but there has been a selective approach to showing improvement on one task while ignoring the other. For example, PointMLP was used as the backbone for classification experiments in Table 3 but ignored for segmentation experiments.

**Questions:**

Although this might sound like a request for more experiments, please include results on a more recent backbone (preferably from 2023 or 2024) to show the relevance/impact on current segmentation methods.

---

> ### Author Response · Authors · 2024-11-19
> **Author Response to Reviewer MMqo**
>
> **Thanks for your time and insightful reviews. We responded in detail as follows:**
>
> **Q1:** The impact of the proposed augmentation techniques has been demonstrated on a limited number of segmentation backbones (precisely PointNet, PointNet++, and DGCNN). These backbones are almost five-year-old methods and have ample room to show the impact. The proposed work does not show its effects in recent methods (PointNeXt onwards), making it unclear if the proposed augmentation approach is relevant to the current state-of-the-art segmentation methods.
> Evaluate
>
> **A1:** In **Table 14, we use more backbones to evaluate the performance of our SinPoint on segmentation task, such as CurveNet (ICCV2021), 3DGCN (TPAMI2021) and GDANet (AAAI2021).  They are the most recent methods.** According to your suggestion, **we further added some of the latest backbones, and we obtained consistent performance improvement. We have uploaded a revision of our paper and marked it in red. The experimental results are as follows:**
>
> **Table R1: Part segmentation performance in the current state-of-the-art segmentation methods.**
> | Method | Baseline | +SinPoint (Ours) |
> |:---|:---:|:---:|
> | PointMLP(ICLR 2022) | 85.8 | **86.1** |
> | SPoTr(CVPR 2023) | 87.0 | **87.2** |
> | PointMetaBase(CVPR 2023) | 86.9 | **87.3** |
> | DeLA(Arxiv 2024) | 87.0 | **87.4** |
>
>
> **Q2:** Figures 8 and 9 in the supplementary pages (page 19) show that the second augmented table generated (using SinPoint SSF) shows that the table top surface has a concave bend so that items placed on it would roll over. Such samples are not realistic or meaningful.
>
> **A2:**
>
> 1.**The purpose of data augmentation is to increase the diversity of training samples so that the model can learn more robust representations. Like Mixup in the field of images, there is no superposition of two images in the real world. Like CutMix, the real world does not have an area mixed with other images. Although these augmented samples don't exist in the real world, they can effectively improve the performance of the model.**
>
> 2.**Our SinPoint is only used during the training phase, when testing, the SinPoint is dropped, only trained backbone model used to inference. Thus, unrealistic or meaningless augmented samples do not affect the models when tested on real samples.**
>
> **Q3:** Inconsistency in using recent models to demonstrate the improvement in classification accuracy for different datasets. For example, PointNeXt, PointMetaBase, and SpoTr are selected to show the improvement in results for the ScanObjectNN dataset's OBJ_ONLY variant in Table 3; the same models are ignored to show the upgrades for the ModelNet40 dataset in Table 2. And vice versa, when considering models in Table 2 that were used to show improvements for the ModelNet40 dataset, they are ignored when showing improvements on ScanObjectNN dataset variants.
>
> **A3:**
>
> 1.Since **the performance of the recent methods is approaching saturation** and is **nearing the theoretical upper bound of the ModelNet40 dataset**. Thus, **the recent methods either failed to outperform PointMLP (94.1) on ModelNet40 or were not tested, such as PointNeXt (93.2), PointMetaBase (not test) and SpoTr (not test)**. **So we chose only some of the most classic backbone networks to validate on ModelNet40.**
>
> 2.**On ScanObjectNN, these new methods have better performance. Therefore, we further validated the performance of our SinPoint on these backbones，such as PointMLP, PointNeXt, PointMetaBase and SpoTr.**
>
> **Q4:** Some of the SOTA models have both classification and segmentation backbones, but there has been a selective approach to showing improvement on one task while ignoring the other. For example, PointMLP was used as the backbone for classification experiments in Table 3 but ignored for segmentation experiments.
>
> **A4: We have added a comparison of segmentation experiments on segmentation task, which we have update in the latest version, and the results as shown in Table R1.**
>
> **Q5:** Although this might sound like a request for more experiments, please include results on a more recent backbone (preferably from 2023 or 2024) to show the relevance/impact on current segmentation methods.
>
> **A5: We have added segmentation backbone experiments, including PointMLP(ICLR 2022), SPoTr (CVPR 2023), PointMetaBase (CVPR 2023), DeLA (Arxiv 2024), and the results as shown in Table R1.**

---

> ### Author Response · Authors · 2024-11-25
> **End of Discussion Approaching**
>
> Dear Reviewer MMqo:
>
> We sincerely appreciate your detailed review and valuable feedback. We have carefully addressed your concerns and made revisions accordingly.
>
> Given that the end of the discussion period is approaching, we would like to ask if there are still any areas that seem unclear or if you have additional questions. We are more than happy to offer additional clarifications.
>
> The current work on point cloud augmentation is largely limited to mimicking image augmentation, while our SinPoint represents a significant innovation and achieves state-of-the-art performance. We hope that these advancements are recognized and valued.
>
> We appreciate the discussion, your feedback, and your suggestions. Many thanks for your time and effort.
>
> Best and sincere wishes,
>
> The authors

---

> ### Author Response · Authors · 2024-11-28
> **Did our answers solve all your questions?**
>
> Dear Reviewer MMqo:
>
> Please allow us to sincerely thank you again for your constructive comments and valuable feedback. We believe our latest response has addressed your points, but please let us know if there is anything else we can clarify or assist with. We are more than happy to answer any further questions during the discussion period. Your feedbacks are truly valued!
>
> Best and sincere wishes,
>
> The authors

---

> ### Author Response · Authors · 2024-12-02
> **Reminder: Rebuttal Discussion for Reviewer MMqo**
>
> As the rebuttal discussion period approaches its end, we kindly remind Reviewer MMqo to review the rebuttal responses addressing all the questions and update the score evaluation for this submission. If the reviewer has any further questions, we would be delighted to provide additional clarification during the remaining discussion period.

---

### Official Review · Reviewer_S1kV · 2024-11-04

**Soundness:** 2
**Presentation:** 2
**Contribution:** 2
**Rating:** 3
**Confidence:** 5

**Summary:**

The paper introduces SinPoint, a novel point cloud data augmentation method that uniquely combines homeomorphism and sine functions. The method addresses two critical challenges: maintaining topological consistency while increasing geometric diversity in point cloud data. SinPoint offers two transformation strategies: Single Sine Function (SinPoint-SSF) and Multiple Sine Function (SinPoint-MSF). The approach demonstrates superior performance on various benchmark datasets, achieving 90.2% accuracy on the ScanObjectNN dataset for shape classification tasks. The method's key innovation lies in its ability to preserve topological structure while simulating natural object deformations, making it versatile and applicable across different point cloud understanding tasks.

**Strengths:**

The method uniquely preserves topological consistency through homeomorphism while creating diverse geometric variations using sine functions, avoiding the common problem of structure distortion that exists in previous point cloud augmentation methods.

**Weaknesses:**

1、This method is based on experiments conducted on older methods such as PointNet++ and DGCNN, and the experimental results are not very convincing because it is well known that these methods can also achieve high performance improvements by adjusting some parameters.
2、The paper lacks comparison with some of the latest point cloud enhancement methods in comparative experiments. At the same time, the stability and robustness analysis of the method under different parameter settings is not sufficient.
3、Although sine functions and homeomorphic mappings are used, the theoretical innovation of the method is not prominent enough. This combination of mathematical tools seems to be relatively intuitive, lacking deeper theoretical insights and mathematical derivations.
4、The article lacks detailed ablation experiments to prove the necessity of each component. In particular, there is a lack of in-depth analysis of the selection basis and performance differences between the SSF and MSF strategies.
5、Can the topological structure information of point clouds be learned through persistent homology? What are the advantages of the method proposed in this paper over persistent homology in maintaining the topological structure of point clouds?
6、The expression of Figure 4 needs to be improved.

**Questions:**

See Weaknesses.

---

> ### Author Response · Authors · 2024-11-19
> **Author Response to Reviewer S1kV (part 1/2)**
>
> **Thanks for your time and insightful reviews. We responded in detail as follows:**
>
> **Q1:** This method is based on experiments conducted on older methods such as PointNet++ and DGCNN, and the experimental results are not very convincing because it is well known that these methods can also achieve high performance improvements by adjusting some parameters.
>
> **A1:** As shown in Table R1, almost all existing point cloud data augmentation efforts use PointNet, PointNet++, and DGCNN as their backbone networks. **To be a fair comparison, we use the same backbones to evaluate our SinPoint method with previous data augmentation methods. Not only that, it can also be seen from Table R1 that in order to verify the stability of SinPoint, we used up to 13 backbones.**
>
> **Table R1: The existing work using backbone:**
> | Method | backbone |
> |:---|:---:|
> | PointAugment (CVPR2020) | PointNet PointNet++ RSCNN DGCNN |
> | PointMixup (ECCV2020) | PointNet PointNet++ |
> | PointWOLF (ICCV2021) | PointNet PointNet++ DGCNN |
> | RSMix (CVPR 2021) | PointNet PointNet++ DGCNN |
> | PatchAugment (ICCVW 2021) | PointNet PointNet++ DGCNN |
> | SageMix (NeurIPS2022) | PointNet PointNet++ DGCNN |
> | WOLFMix (PMLR 2022) | PointNet PointNet++ DGCNN PCT GDANet |
> | PCSalMix (ICASSP 2023) | PointNet PointNet++ DGCNN |
> | PointPatchMix (AAAI2024) | PointNet PointNet++ Transformer PointMLP PointMAE |
> | SinPoint (Ours) | **PointNet PointNet++ DGCNN RSCNN PointConv PointCNN GDANet PCT CurveNet 3DGCN PointMLP(2022) PointNeXt(2022) PointMetaBase(2023) SPoTr(2023)** |
>
>
> **At the same time, we acknowledge that parameter tuning can improve model accuracy. However, as shown in Table R2, the performance gains of up to 6.7, 5.1, and 7.3 in PB_T50_RS cannot be achieved through tuning alone.**
>
> **Table R2: Our SinPoint evaluate on PB_T50_RS.**
> | PB_T50_RS | Backbone | +SinPoint (Ours) |
> |:---|:---:|:---:|
> | PointNet | 64.1 | **70.8 (+6.7)** |
> | PointNet++ | 79.4 | **84.5 (+5.1)** |
> | DGCNN | 77.3 | **84.6 (+7.3)** |
>
> **Q2:** The paper lacks comparison with some of the latest point cloud augmentation methods in comparative experiments.
>
> **A2:** Thanks for your suggestion. **We have added two mix-based methods in Table R3, and it is clear that our SinPoint outperforms the latest methods.** We will include this update in the next version. **We have uploaded a revision of our paper and marked it in yellow.**
>
> **Table R3: New contrast.**
> | Method | PointNet | PointNet++ | DGCNN |
> |:---|:---:|:---:|:---:|
> | PCSalMix(ICASSP2023) | 90.48 | 93.11 | 93.19 |
> | PointPatchMix(AAAI2024) | 90.1 | 92.9 | – |
> | **SinPoint (Ours)** | **91.3** | **93.4** | **93.7** |
>
> **Q3:** At the same time, the stability and robustness analysis of the method under different parameter settings is not sufficient.
>
> **A3: Our ablation studies are complete, and we have conducted a detailed stability and robustness analysis along with ablation experiments using different parameters.**
>
> 1.In Table 4, we study the robustness of our model under different perturbations, going beyond the optimal approach.
>
> 2.In Table 5, we conducted a detailed ablation study of different strategy augmentation.
>
> 3.In Table 8, we conducted an ablation study on the stability of the augmentation method.
>
> 4.In Table 9, the ablation of the parameter angular velocity w is studied.
>
> 5.In Table 10, we conducted ablation studies on the parameter amplitude A.
>
> 6.In Table 11, we conducted an ablation study on the number of parametric anchor points k.
>
> 7.In Table 12, we even compare the classification performance differences between the two strategies.
>
> 8.In Table 13, we conducted an ablation study of sampling strategies for parameters w and A.
>
> **Q4:** Although sine functions and homeomorphic mappings are used, the theoretical innovation of the method is not prominent enough. This combination of mathematical tools seems to be relatively intuitive, lacking deeper theoretical insights and mathematical derivations.
>
> **A4:** Thanks for your suggestion, we think our innovation is obvious. Here are the facts:
>
> 1.**Our SinPoint is the first attempt to consider using homeomorphic mapping and sine function for point cloud augmentation.** Our SinPoint ensures that the augmented point cloud maintains topological consistency while increasing geometric diversity. In addition, our SinPoint is simple and easy to follow.
>
> 2.**Although the use of the sine function may seems to be relatively intuitive, but it has not been previously applied to point cloud augmentation. This novelty makes our method distinct in the point cloud augmentation task.**
>
> 3.**Although sine functions and homeomorphic mappings are well known, but this is the first time we have proposed such a combination and it has been rigorously proved theoretically (Theorem 1). Therefore, our work is not just a simple application of mathematical tools, but an innovative contribution based on in-depth theoretical analysis.**

---

> ### Author Response · Authors · 2024-11-19
> **Author Response to Reviewer S1kV (part 2/2)**
>
> **Q5:** The article lacks detailed ablation experiments to prove the necessity of each component.
>
> **A5:** In **page 5, we conducted a module ablation studies and gave a detailed analysis. The experimental results are shown in Table 5.**
>
> **Q6:** In particular, there is a lack of in-depth analysis of the selection basis and performance differences between the SSF and MSF strategies.
>
> **A6:** First, **we present the SSF and MSF visualization comparison in Figure 2. The SSF has a single change, while the MSF has a variety of changes.** Second, **in Table 6 we perform ablation experiments of SSF and MSF on segmentation, and we analyze why MSF is more suitable for segmentation.** Finally, **in Tables 11 and 12, we present the ablation comparison of SSF and MSF on classification.**
>
> **To summarize:**
>
> 1.**SinPoint-MSF is more suitable for point cloud segmentation tasks,** because it enhances the sensitivity of the model to different scales, morphology and local details through diverse deformation methods, and can better deal with complex region boundaries and shape changes, thereby improving the accuracy of point cloud segmentation.
>
> 2.**SinPoint-SSF is more suitable for classification tasks,** because it simplifies the deformation process, focuses on the learning of global features, avoids the interference of local details, and can be more efficient for category discrimination, especially in classification tasks with high efficiency and accuracy.
>
> **Q7:** Can the topological structure information of point clouds be learned through persistent homology? What are the advantages of the method proposed in this paper over persistent homology in maintaining the topological structure of point clouds?
>
> **A7:** Here we see **a misunderstanding**. **Persistent homology is obviously different from Homeomorphism, as shown in Table R4.**
>
> **Table R4: The relationship and comparison between Homology and Homeomorphism**
> | Property | Homology | Homeomorphism |
> |:---|:---:|:---:|
> | Definition | A homology group describes holes, connected components of a space | A bijective and continuous map has a continuous inverse map |
> | Characteristic | Only topological features are described, and geometric and differential information is ignored | The topology is identical |
> | Transformations | Tearing or merging is allowed as long as homogeneity is maintained | Continuous deformation is allowed, no tearing or merging is allowed |
> | Application | Topology data analysis, shape recognition | Spatial classification in topology |
>
> From table R4 we observe that:
>
> 1.Persistent homology and homeomorphism are two completely different concepts in topology, and they have different emphases and application scenarios in the study of spatial shape and structure.
>
> 2.Persistent homology provides a way to quantify and analyze the topology of data, while homeomorphism is a more abstract equivalence relation for comparing topological properties of different Spaces.
>
> 3.**Our SinPoint is not learning topologies. It is just using topological consistency to construct diverse samples. Refer to Figure 4 and Algorithm 1.**
>
> **Q8:** The expression of Figure 4 needs to be improved.
>
> **A8:** Thanks for your suggestion, which is important for us to improve the quality of our drawings. **We have updated it in the latest version.**

---

> ### Author Response · Authors · 2024-11-25
> **End of Discussion Approaching**
>
> Dear Reviewer S1kV:
>
> We sincerely appreciate your detailed review and valuable feedback. We have carefully addressed your concerns and made revisions accordingly.
>
> Given that the end of the discussion period is approaching, we would like to ask if there are still any areas that seem unclear or if you have additional questions. We are more than happy to offer additional clarifications.
>
> The current work on point cloud augmentation is largely limited to mimicking image augmentation, while our SinPoint represents a significant innovation and achieves state-of-the-art performance. We hope that these advancements are recognized and valued.
>
> We appreciate the discussion, your feedback, and your suggestions. Many thanks for your time and effort.
>
> Best and sincere wishes,
>
> The authors

---

> ### Author Response · Authors · 2024-11-28
> **Did our answers solve all your questions?**
>
> Dear Reviewer S1kV:
>
> Please allow us to sincerely thank you again for your constructive comments and valuable feedback. We believe our latest response has addressed your points, but please let us know if there is anything else we can clarify or assist with. We are more than happy to answer any further questions during the discussion period. Your feedbacks are truly valued!
>
> Best and sincere wishes,
>
> The authors

---

> ### Author Response · Authors · 2024-12-01
> **End of Discussion Approaching**
>
> Dear Reviewer S1kV:
>
> We sincerely appreciate your detailed review. We have carefully addressed your concerns and made revisions accordingly.
>
> Given that the end of the discussion period is approaching, we would like to ask if there are still any areas that seem unclear or if you have additional questions. We are more than happy to offer additional clarifications.
>
> Best and sincere wishes,
>
> The authors

---

> ### Author Response · Authors · 2024-12-02
> **Reminder: Rebuttal Discussion for Reviewer S1kV**
>
> As the rebuttal discussion period approaches its end, we kindly remind Reviewer S1kV to review the rebuttal responses addressing all the questions and update the score evaluation for this submission. If the reviewer has any further questions, we would be delighted to provide additional clarification during the remaining discussion period.

---

> ### Comment · Reviewer_S1kV · 2024-12-02
>
> Dear author,
>
> I have discussed your question in the comments above and shared my thoughts on your paper and the problems in this subfield.
>
> Other Comments:
>
> - If you were to separately apply translation, rotation, and random drop operations to the input data, you might achieve better experimental results. The reason is that both approaches involve scaling up the dataset size by a factor of n, which in my opinion is not particularly innovative. Even though you have linked the concept of homeomorphism to the sine function, this remains your only innovation, with no other significant contributions.
>
> - There is another question. You have been emphasizing local homeomorphism and global homeomorphism in the article, which makes it easy for people to think that local homeomorphism will lead to global homeomorphism. In fact, the two are not equivalent. You did not emphasize this point in the article.
>
> - Additionally, I noticed that your experiments are based on the PointMLP GitHub repository. However, in the code you provided, the `utils` folder is missing a subdirectory named `progress`. This file needs to be copied from the PointMLP repository for your program to run properly.
>
>
> -**Most importantly, although the pre-trained model you provided has a test accuracy of 90.017% on my machine, I cannot reproduce the accuracy you reported on my two different types of GPUs. And the accuracy I reproduced (about 86%) is too different from the accuracy you reported (90.2%). Therefore, the credibility of your method and its superiority over other methods are questionable.**
>
>
> Best Regards,
> Reviewer S1kV

---

> > ### Author Response · Authors · 2024-12-03
> > **Sincerely respond to Reviewer S1kV's comments**
> >
> > Dear Reviewer S1kV:
> >
> > Q1 If you were to separately apply translation, rotation, and random drop operations to the input data, you might achieve better experimental results. The reason is that both approaches involve scaling up the dataset size by a factor of n, which in my opinion is not particularly innovative. Even though you have linked the concept of homeomorphism to the sine function, this remains your only innovation, with no other significant contributions.
> >
> > A1 **The method you said is too engineering and has no theoretical basis. PointWOLF is that kind of work, and we outperform it.**
> >
> > Q2 There is another question. You have been emphasizing local homeomorphism and global homeomorphism in the article, which makes it easy for people to think that local homeomorphism will lead to global homeomorphism. In fact, the two are not equivalent. You did not emphasize this point in the article.
> >
> > A2 **Refer to Figure 2.**
> >
> > Q3 Additionally, I noticed that your experiments are based on the PointMLP GitHub repository. However, in the code you provided, the utils folder is missing a subdirectory named progress. This file needs to be copied from the PointMLP repository for your program to run properly.
> >
> > A3 **Thanks for reminding us; we will provide it in the subsequent github library.**
> >
> > Q4 Most importantly, although the pre-trained model you provided has a test accuracy of 90.017% on my machine, I cannot reproduce the accuracy you reported on my two different types of GPUs. And the accuracy I reproduced (about 86%) is too different from the accuracy you reported (90.2%). Therefore, the credibility of your method and its superiority over other methods are questionable.
> >
> > A4 **All of our experiments were conducted on NVIDIA TITAN RTX GPUs, and as such, we did not specifically adapt our code to other GPUs. We have observed that performance can vary across different GPU+Torch+CUDA versions. We plan to address this in the final version by ensuring compatibility and optimizing performance for a broader range of GPUs.**
> >
> > On RTX 3090
> >
> > Vanilla out: {'loss': 1.293, 'acc': 90.017, 'acc_avg': 88.428, 'time': 3}
> >
> > On RTX 4090
> >
> > Vanilla out:{'loss':1.292,'acc':90.189,'acc avg':88.547,'time': 1}
> >
> > On NVIDIA TITAN RTX
> >
> > Vanilla out: {'loss': 1.292, 'acc': 90.189, 'acc_avg': 88.666, 'time': 5}
> >
> > Best and sincere wishes,
> >
> > The authors

---

> > > ### Comment · Reviewer_S1kV · 2024-12-03
> > >
> > > Dear author,
> > >
> > > Thank you for your previous response to Q1-Q6 of the "Weaknesses" section. I appreciate your efforts and work in the rebuttal stage. I wanted to change my score before, but after re-reviewing your paper and code, I have a different opinion of your paper. Your code provides a negative impact on your paper.
> > >
> > > As a professional in the field of point cloud analysis, I quickly saw that your code is based on the PointMLP github code base, and I also know which dependency files your code is missing. So I am familiar with the field of point clouds. According to the configuration you provided, I quickly ran through your code without making any changes. Even though different types of GPUs and configurations will produce different effects, the experimental results I reproduced on two types of GPUs show that the performance you reported is far from what you reported. In addition, for the latest Q2, you asked me to refer to Figure 2, but Figure 2 is just a visualization of the results, without other direct information.
> > >
> > > Therefore, you did not fully address the core concerns I raised. I will continue to maintain my opinion and current score.
> > >
> > > Best Regards,
> > > Reviewer S1kV

---

> ### Author Response · Authors · 2024-12-04
> **Sincerely respond to Reviewer S1kV**
>
> Dear Reviewer S1kV,
>
> **Thanks for your responses. I appreciate the effort and time you have put into the rebuttal stage.**
>
> Q1: To be brief, as a professional in the field of point cloud analysis, I quickly noticed that your code is based on the PointMLP GitHub repository, and I am also aware of which dependency files are missing from your code.
>
> A1: **This issue is related to a missing public library, and our core code remains intact.**
>
> Q2: Based on the configurations you provided, I reproduced your experimental results on two types of GPUs. However, the performance I obtained differs significantly from the results you reported. Although different types of GPUs and configurations may lead to varying outcomes, your reasoning that my training process overfitted seems questionable.
>
> A2: **We have already answered this question under another question** (Question 1: but after re-reviewing your paper and code, I have a different opinion of your paper. Your code provides a negative impact on your paper).
>
> Q3: I followed the information you provided for training, and since this issue occurred on both types of GPUs, doesn’t this indicate that your code lacks adaptability?
>
> A3: **The outcomes you obtained align closely with overfitting. We encourage you to try the experiment again with the necessary adjustments, and we are confident that you will achieve the desired results.**
>
> Q4: I believe the core concern lies in your doubling of the data, which enlarged the training set and caused the model to overfit on it. This appears to be a critical issue, and your code seems to have had a negative effect.
>
> A4: **It is well-established that data augmentation helps reduce the risk of overfitting. We have already answered this question under another question** (Question1: but after re-reviewing your paper and code, I have a different opinion of your paper. Your code provides a negative impact on your paper).
>
> Q5: At the same time, you misunderstood the parallel operation of data enhancement I mentioned. The parallel operation here is not the parallel operation on the GPU, but you will get triple data after data enhancement is translated, rotated, and randomly dropped. This operation triples the data set. Your method is similar to this operation, and you get two tuples.
>
> A5: **This issue appears to be unrelated to our method. The concern remains unclear to us, as we do not perform the operation you mentioned. You can verify this by reviewing our code.**
>
> Q6: A minor suggestion: In future work, I suggest validating your data augmentation method in downstream fine-tuning tasks under the current trending self-supervised models, rather than retraining during the pretraining phase.
>
> A6: **Thanks for your suggestion, which we believe is worth exploring in future work. However, our current focus is on supervised learning, and we consider this aspect of the work to be complete at this stage.**
>
> Q7: Additionally, it is worth emphasizing that the backbone of PointGPT is approximately 22M in size. When you mentioned that the parameters of PointGPT are around 300M, you likely included other branches from the pretraining process.
>
> A7: **The full parameter count of PointGPT is 300M, but we noticed that you mentioned using only 22M, which may have been a misunderstanding. However, even with 22M parameters, we were able to achieve comparable results by utilizing just 1.7M parameters and a smaller training dataset. Meanwhile, this discussion may not be directly relevant to our paper.**
>
> **In conclusion, I thank you again for your efforts and responses. We wish the reviewers good health and a speedy recovery.**
>
> Best regards,
>
> The authors

---

### Author Response · Authors · 2024-11-30
**Global response to all reviewers**

Dear Reviewers,

I would like to take this opportunity to present our research work in detail and outline the key contributions of our study.

**Contributions:**

**1 Innovative Topology Consistency Augmentation Strategy:**

We propose a novel point cloud data augmentation method based on homeomorphic mapping. Through extensive experimental validation, our method, SinPoint, achieves state-of-the-art performance in the field of point cloud augmentation.

**2 Pioneering a New Path for Point Cloud Augmentation:**

Our approach opens new research avenues for point cloud augmentation, which does not rely on the conventional framework of simulating image mixing for data augmentation. It outperforms all existing mix-based methods in terms of performance.

**3 Solving the Problem of Topology Destruction:**

In contrast to methods like PointWOLF, our SinPoint effectively avoids the destruction of point cloud topology during augmentation while simultaneously increasing geometric diversity and ensuring both topological and semantic consistency.

**4 Theoretical Contribution:**

We introduce a sine function-based homeomorphic mapping for point cloud augmentation, with detailed theoretical proof provided.

**Experimental Verification:**

**1 Extensive Experiments on Synthetic and Real-World Datasets:**

We conduct extensive experiments on synthetic datasets such as ModelNet40 and ShapeNetPart, as well as real-world datasets like ScanObjectNN, to compare our method with existing approaches. Our experiments demonstrate that SinPoint achieves state-of-the-art performance.

**2 Comprehensive Evaluation Across Multiple Backbones:**

To verify the general applicability of our method, we evaluate it using 13 different backbone networks, showing the robustness and versatility of SinPoint compared to other point cloud augmentation algorithms.

**3 Ablation Studies:**

We also conduct numerous ablation experiments to demonstrate the stability and robustness of our method under various conditions.

**4 Expand scene segmentation:**

On the S3DIS and SemanticKITTI datasets, our SinPoint can still effectively enhance the performance of backbone.

**Application Prospects:**

**1 New Ideas for Point Cloud Augmentation:**

Our work introduces a new perspective on point cloud augmentation based on deformation function, providing researchers with an approach that aligns more closely with the intrinsic properties of point cloud data.

**2 Advancing Point Cloud Contrastive Learning:**

In the domain of point cloud contrastive learning, the construction of positive samples is crucial. While current methods like CDA (e.g., scaling, rotation, and translation) and PointWOLF can generate positive samples, CDA suffers from insufficient diversity, and PointWOLF fails to maintain topological and semantic consistency. Our method offers a significant improvement in constructing diverse and consistent positive samples, making it highly promising for advancing contrastive learning in point clouds.

**Thank you for your time and attention in reviewing our work.**

**We hope that our work can be recognized and valued by all reviewers.**

Sincerely,

The Authors

---

### Author Response · Authors · 2024-12-02
**Reviewer S1kV raised many questions that were not consistent with the facts**

Dear PC, SAC, AC

**We disagree with the reviewer S1kV. The reviewer S1kV is extremely irresponsible and did not read our paper carefully. Reviewer S1kV raised many questions that were not consistent with the facts and did not participate in the discussion.**

The **evidence** is as follows (**Q1 to Q7 come from our splitting of the reviewer question in our rebuttal**):

**1) The review comments Q1 provided by reviewer S1kV indicate that the reviewer did not review the paper carefully, and the questions raised were all subjective assumptions without supporting evidence.**

It is obvious that reviewer S1kV lacks relevant background and has not fully reviewed our paper. Currently, all point cloud augmentation methods rely on a maximum of 5 backbones, whereas we have used 13 backbones for extensive experimental verification. In our rebuttal, we provide a detailed summary of the existing work, which typically employs PointNet, PointNet++, and DGCNN as backbones. In contrast, we use 13 distinct backbones for evaluation, but the reviewer seems to have overlooked this.

In addition, reviewer S1kV said baseline tuning also improves performance. But how much can be improved, which papers have experimental proof, he did not say. The reviewer also did not provide corresponding literature, which left us a little confused.

**2) The review comments Q1 and Q2 provided by reviewer S1kV are contradictory.**

Reviewer S1kV reminded us to ignore two recent studies in point cloud augmentation. However, these methods are still only using PointNet, PointNet++ and DGCNN as the backbone. ** This shows that the reviewers were well aware of the experimental setup for point cloud expansion but still proposed Q1, ignoring our experiments using up to 13 different bone networks. **

We have now included relevant comparisons in the latest version of our paper, and our SinPoint still achieves SOTA performance.

**3) The review comments Q3, Q5 and Q6 provided by reviewer S1kV are irresponsible.**

**The reviewer jumped to conclusions without carefully reading the paper, which is also perfunctory and irresponsible.** We conducted detailed ablation experiments on various parameters. Up to eight groups of ablation experiments were performed. **In the rebuttal, we provided a detailed list of these experiments, and we also highlighted the specific locations of the experiments within the paper for the reviewer's reference.**

**3) The review comments Q4 provided by reviewer S1kV are unreasonable.**

We claim that we made an innovation contribution, which is also supported by other reviewers.

As reviewer 2azz said, this paper presents a novel data augmentation method called SinPoint, which enhances the geometric diversity of point clouds while preserving the topological structure of the original point cloud through a homeomorphism.

As reviewer pDnW said, the paper introduces a data augmentation method for point cloud understanding that utilizes the sine function to generate smooth displacements and preserve the topological structure of the original points via a homeomorphism. The authors propose a simple but effective data augmentation method in some scenarios.

As reviewer MMqo said, the paper proposes a novel point cloud data augmentation technique called SinPoint. It is based on a homomorphism and maintains topological consistency while increasing the geometric diversity of samples (perturbs the local structure using a sine function).

**It is unreasonable for reviewer S1kV to reject our work for lack of innovation. We are the first to propose a method for augmenting point clouds that is based on topological consistency. Our SinPoint is supported by detailed proof and experimental verification.**

**4) The review comments Q7 provided by reviewer S1kV have nothing to do with our paper.**

The reviewer S1kV mentioned "persistent homology" and asked us to compare it with our method. These are two mathematical concepts with similar names but distinct differences. The reviewer also did not provide corresponding references, which left us a little confused. Nevertheless, we researched and provided a comparison. However, "persistent homology" has nothing to do with our work.

**Additionally, we have provided detailed responses to reviewer S1kV's questions and have included supplementary explanations regarding the contributions and significance of our work in our replies to other reviewers. Many experiments have been conducted to demonstrate the effectiveness and innovation of our SinPoint.**

**We sincerely hope that PC, SAC and AC will consider the innovation of our work and our contribution to point cloud augmentation.**

**Meanwhile, we hope PC, SAC and AC can double check the qualification of reviewer S1kV.**

Thanks a lot!

Best and sincere wishes,

The authors of Paper 5358

---

> ### Comment · Reviewer_S1kV · 2024-12-02
>
> Thank you for the author's reply. I stopped all work due to health reasons some time ago. As a researcher in the field of point cloud analysis, I am confident in my professionalism. I have been reviewing your paper today, and I will reply to the latest other replies within five hours.

---

> > ### Author Response · Authors · 2024-12-02
> > **Author Response to Reviewer S1kV**
> >
> > Dear Reviewer S1kV:
> >
> > Thank you for your update, and I hope you're feeling better. I completely understand the need to prioritize health, and I appreciate your dedication to reviewing the paper despite these challenges.
> >
> > It's great to hear that you're confident in your expertise in point cloud analysis; then I'm sure you're familiar with all the point cloud augmentation works I've compared. I'll be waiting for your feedback for the next five hours.
> >
> > Wishing you a speedy recovery and good health.
> >
> > Best regards,
> >
> > The authors

---

> ### Comment · Reviewer_S1kV · 2024-12-02
> **Point cloud data augmentation methods use double the data, making it impossible to validate the method's effectiveness, despite wrapping the method in topological homeomorphism concepts. Additionally, results cannot be reproduced based on the code.**
>
> Dear PC, SAC, AC and Authors:
>
> After careful review and based on my years of experience in point cloud analysis, this paper still fails to meet acceptable standards. The authors' complaints primarily derive from the "Review Feedback by Associate Program Chairs" suggestions rather than addressing my fundamental concerns, which stem from actual research experience and empirical validation.
>
> 1. **Regarding their first complaint:**
>
> While existing methods utilize PointNet, PointNet++ and DGCNN backbones (pre-2019), I expected to see evaluations on current (2022-2023) methods such as RepSurf or self-supervised models (ReCon, ACT) for downstream tasks. PointNet, PointNet++ and DGCNN have relatively low performance baselines, making it easier to inflate performance improvements. While Table 1 shows relative gains, Table 2 reveals only 0.2-0.3% improvements over PCT and PointMLP - insufficient evidence of method effectiveness. Although the authors used some 2022 and 2023 methods in certain experiments, the performance improvement is weak. Although there is an improvement, the main reason is that **they use double the data**.
>
> 2. **Regarding their second complaint:**
>
> The authors claim I lack background knowledge in one complaint but state I am well-versed in point cloud augmentation in another - a contradiction of their own making. My concern isn't about the number of backbones tested, but whether Their method incorporates the latest backbones. Although the authors used some 2022 and 2023 methods in certain experiments, their improvements are weak. Although their method is slightly improved, **they used twice the data, which is an unfair comparison. I will provide evidence below**. As a reviewer, my goal is to advance the field rather than perpetuate outdated experimental protocols, especially given self-supervised learning's dominance in modern object-level point cloud analysis.
>
> 3. **Regarding their third complaint:**
>
> The amplitude ablation study (Table 10) shows 91.189% accuracy at A=0.6 on OBJ_ONLY but only around 88.9% at A=0.2, 0.4, and 1.0 - comparable to PointWOLF. Although the authors wrap the method with topomorphism, this drastic change undermines confidence in the method.
>
> 4. **Regarding their innovation claims:**
>
> Upon code review, the method is remarkably simple - applying sinusoidal coordinate mapping and adding it to the original coordinates. This single technical contribution is wrapped in topological homeomorphism theory but lacks substantive innovation. Even if the author would refute that other methods also do this, for me who is familiar with the field of point clouds, I think this method does not meet ICLR's requirements for innovation.
>
>
> 5. **Regarding persistent homology:**
>
>
> This question arose from their discussion of local homeomorphisms. Persistent homology also analyzes object topological structure information from a topological perspective, extracting global topological information between homeomorphic objects, and has established precedent in point cloud analysis, as evidenced by "Adaptive Topological Feature via Persistent Homology: Filtration Learning for Point Clouds" (NIPS 2023) and "Persistent Homology based Graph Convolution Network for Fine-grained 3D Shape Segmentation" (ICCV 2021). Therefore, this is an open problem related to point cloud topological transformation, rather than an unrelated problem.
>
>
> While I regret not responding promptly to the authors, your approach did not convince me. In my following response, I will provide evidence supporting why I chose to reject this manuscript.
>
> Best Regards,
>
>
> Reviewer S1kV

---

> > ### Comment · Reviewer_S1kV · 2024-12-02
> >
> > Dear PC, SAC, AC, other Reviewers, and Authors,
> >
> > I am now providing the reasons why I have rated this manuscript poorly. After carefully reviewing the code and running it on two different types of GPUs, I have identified the following issues:
> >
> > ---
> >
> > ### **1. Inability to reproduce the experimental results claimed in the manuscript.**
> >
> > (1). **On RTX 4090:**
> >    The reproduced result was **85.886%**:
> >    - The best model trained achieved the following results on the validation set (also used as the test set):
> >
> >      ```
> >      ++++++++++++++++Final results++++++++++++++++
> >      ++  Last Train time: 14 | Last Test time: 1  ++
> >      ++  Best Train loss: 1.097 | Best Test loss: 1.352  ++
> >      ++  Best Train acc_B: 99.577 | Best Test acc_B: 84.322  ++
> >      ++  Best Train acc: 99.631 | Best Test acc: 85.886  ++
> >      ++++++++++++++++++++++++++++++++++++++++
> >      ```
> >
> >    - Test result:
> >      ```
> >      Vanilla out: {'loss': 1.367, 'acc': 85.886, 'acc_avg': 84.155, 'time': 1}
> >      ```
> >
> > (2). **On RTX 5000:**
> >    The reproduced result was **86.059%**:
> >    - The best model trained achieved the following results on the validation set (also used as the test set):
> >
> >      ```
> >      ++++++++++++++++Final results++++++++++++++++
> >      ++  Last Train time: 31 | Last Test time: 3  ++
> >      ++  Best Train loss: 1.12 | Best Test loss: 1.353  ++
> >      ++  Best Train acc_B: 99.321 | Best Test acc_B: 83.134  ++
> >      ++  Best Train acc: 99.306 | Best Test acc: 86.059  ++
> >      ++++++++++++++++++++++++++++++++++++++++
> >      ```
> >
> >    - Test result:
> >      ```
> >      Vanilla out: {'loss': 1.373, 'acc': 86.059, 'acc_avg': 83.134, 'time': 18}
> >      ```
> >
> >    The authors claimed an accuracy of **90.2%** on the OBJ ONLY dataset. Although the GPUs used are not identical to those of the authors, the results exhibit a significant discrepancy (4%), which raises substantial doubts about the manuscript's validity and reproducibility.
> >
> > ---
> >
> > ### **2. Double the training data was used in the experiments.**
> >
> > In the released code, the authors concatenate the sine-transformed data with the original data (via `torch.cat`), effectively doubling the training data. This is unfair when comparing to other backbones. The relevant code is as follows:
> >
> > ```python
> > def Sin(self, data, label=[]):
> >     """
> >     Args:
> >         data (B,N,3)
> >         label (B)
> >     """
> >     B, _, _ = data.shape
> >     newdata, shift, scale = self.normalize_point_clouds(data)
> >     if self.rand_center_num == 0:
> >         newdata = self.Global(newdata)
> >     else:
> >         newdata = self.Local(newdata)
> >     newdata = newdata * scale + shift
> >     label = label.unsqueeze(1)
> >     if self.isCat:
> >         newdata = torch.cat([data, newdata], dim=0)
> >         label = torch.cat([label, label], dim=0)
> >         if self.shuffle:
> >             idxs = torch.randperm(B*2)
> >             newdata = newdata[idxs, :, :]
> >             label = label[idxs, :]
> >     return newdata, label.squeeze(1)
> > ```
> >
> > Here, `self.Global` and `self.Local` apply different sine transformations. This concatenation of original and transformed data is the only innovation in the paper. Can such a simple innovation, wrapped in topological homeomorphism theory, meet the standards of ICLR?
> >
> > ---
> >
> > ### **3. Unfair Comparisons**
> >
> > (1) In response to possible rebuttals from reviewers that other point cloud data augmentation methods also double the dataset size, I want to emphasize that your experiments are unfair. For instance, in Table 3, you only improve PointMLP from 94.1% to 94.3%, a mere 0.2% increase. However, you used twice the training data, whereas PointMLP only utilized the original dataset. Is such a comparison fair?
> >
> > (2) Your method is also unfair compared to other similar point cloud data augmentation methods, such as PointWOLF (ICCV 2021). PointWOLF transforms the local coordinates of the point cloud without applying translation and then optimizes the model by summing the losses from the original and transformed data separately. Even though PointWOLF also uses double the training data, your approach differs because you apply translation before the sine transformation. You concatenate the translated point cloud with the one that underwent both translation and sine transformation. Do you believe this is a fair comparison? How is this different from turning serial augmentation into parallel augmentation?
> >
> > Additionally, PointWOLF uses 250 epochs, whereas your method employs 300 epochs. Isn't this a parameter discrepancy?
> >
> > ---

---

> > > ### Comment · Reviewer_S1kV · 2024-12-02
> > >
> > > ### **4. Reviewer's Personal Research Experience:**
> > >
> > > Before encountering your method, I had already studied the impact of point cloud data augmentation on model training. Existing point cloud augmentations process the data serially, which is what you refer to as traditional data augmentation (CDA) in your paper — passing the point cloud through scaling, rotation, and translation in sequence. However, this approach has a limitation: it cannot ensure data diversity. Parallel data augmentation can increase training samples by applying scaling, rotation, and translation separately, expanding the training data threefold. Each set of data is then passed through the model independently, and their losses are summed for optimization. Using this method, I improved the ScanObjectNN accuracy of PointGPT-S by at least 3%.
> > >
> > > However, your method lacks comparisons with self-supervised methods. This is why I suggested testing your approach against state-of-the-art methods such as ACT and ReCon. You repeatedly highlight that other augmentation methods were tested only on PointNet, PointNet++, and DGCNN, which were proposed before 2019. If your method is generalizable, it should be tested on the latest pre-trained point cloud models under different learning paradigms to truly advance the field. Meanwhile, your augmentation process — applying translation first, then combining translated and sine-transformed data —may be invalid，specifically refers to the sine transformation. Even without your sine transformation, simply applying translation and rotation separately, and then feeding them into the network, could achieve similar or better results. Furthermore, PointMLP only uses the original dataset, whereas you double the data size. Thus, you should explicitly acknowledge the unfairness in the point cloud data augmentation field rather than perpetuate it.
> > >
> > > ### **5. Minor Issues**
> > >
> > > In Table 1, the authors do not clarify whether SinPoint-SSF or SinPoint-MSF was used. Although this is mentioned in the appendix, the results for these two variations should be included in Table 1 instead of ambiguously referring to both as "SinPoint."
> > >
> > > ---
> > >
> > > ### **Conclusion**
> > >
> > > In summary, I hope the authors refrain from justifying their approach by citing other point cloud data augmentation methods that do similar things. What I want to see is an effort to address the fairness issues in this subfield! In my opinion, no matter how well the paper dresses its method up with topological homeomorphism theory, it essentially boils down to applying a translation followed by a sine transformation and doubling the input data. The core contribution of the paper is limited to this single step, and it raises significant concerns about the novelty and fairness of comparisons.
> > >
> > > Best Regards,
> > > Reviewer S1kV

---

> > > ### Author Response · Authors · 2024-12-03
> > > **Sincerely respond to Reviewer S1kV's comments**
> > >
> > > Dear PC, SAC, AC , other Reviewers, and Reviewer S1kV,
> > >
> > > We completely disagree with the Reviewer S1kV, and the evidence is as follows:
> > >
> > > 1. Inability to reproduce the experimental results claimed in the manuscript.
> > >
> > > **Table K1: The Reviewer S1kV reproduces the results.**
> > > | Reviewer S1kV | Baseline |
> > > |:---|:---:|
> > > | On RTX 4090 85.886 | 86.2 |
> > > | On RTX 5000  86.059 | 86.2 |
> > >
> > > **The results of the Reviewer S1kV did not even reach the baseline, and we seriously doubt the correctness of the experiment.**
> > >
> > > **We provide the complete training process, code, parameters, and pre-training model in the supplementary material.**
> > >
> > > We train on NVIDIA TITAN RTX:
> > >
> > > **The best model trained achieved the following results on the validation set (also used as the test set):**
> > >
> > > ++++++++++++++++Final results++++++++++++++++
> > >
> > > ++  Last Train time: 30 | Last Test time: 1  ++
> > >
> > > ++  Best Train loss: 1.225 | Best Test loss: 1.282  ++
> > >
> > > ++  Best Train acc_B: 93.477 | Best Test acc_B: 88.666  ++
> > >
> > > ++  Best Train acc: 93.945 | Best Test acc: 90.189  ++
> > >
> > > ++++++++++++++++++++++++++++++++++++++++
> > >
> > > Test result:
> > >
> > > Vanilla out: {'loss': 1.292, 'acc': 90.189, 'acc_avg': 88.666, 'time': 5}
> > >
> > > **At the same time, we found that the reviewer's training process had been overfit, and the training set reached.**
> > >
> > > Best Train acc_B: 99.577
> > >
> > > Best Train acc: 99.631
> > >
> > > On the training set, we only:
> > >
> > > Best Train acc_B: 93.477
> > >
> > > Best Train acc: 93.945
> > >
> > > **This further proves that the reviewer's experiment is seriously flawed. It is recommended that reviewers refer to Table5 in line 417 and A IMPLEMENTATION DETAIL in Line 710 for experimental replication.**
> > >
> > > Reviewer S1kV was unable to reproduce our results, possibly due to differences in GPU hardware or environmental setup, or because the version of our code in the initial submission was outdated. In the final version, we will provide the GitHub link and upload the updated, cleaned code. We will thoroughly test across all GPUs and CUDA environments to ensure reproducibility in a variety of setups.
> > >
> > > 2. Double the training data was used in the experiments.
> > >
> > > 1) **Thank you for thoroughly reviewing our code. Your detailed examination helps to demonstrate that our implementation is consistent with the results presented in Figure 4. The goal of self-augmentation is to enhance geometric diversity, and by using both raw and augmented data, the model is able to learn a broader range of geometric features. This approach is fundamentally different from sample mixing methods. For a fair comparison, our method should be compared with other point cloud augmentation techniques rather than with backbone architectures.**
> > >
> > > 2) **We would like to reiterate that the core innovation of our work lies in proposing a point cloud augmentation algorithm based on homeomorphic mapping. We have theoretically justified the rationale behind our method and validated its effectiveness through extensive experiments. With both theoretical foundations and empirical evidence, we are confident that our approach fully meets the standards set by ICLR.**
> > >
> > > 3) **Similar to ResNet, which addresses the vanishing gradient problem with a simple residual link that requires just a single line of code. Our approach uses only a sine function to ensure topological consistency. This simplicity enables point cloud augmentation to achieve state-of-the-art performance.**

---

> > > ### Author Response · Authors · 2024-12-03
> > > **Sincerely respond to Reviewer S1kV's comments**
> > >
> > > 3. Unfair Comparisons
> > >
> > > 1) We would like to emphasize once again that our work focuses on data augmentation, and we must use the all dataset for training. The goal of our experiment is to demonstrate that increasing the diversity of training examples can improve model performance. As Reviewer S1kV pointed out, PointMLP uses only raw data, while PointMLP+SinPoint utilizes both raw and augmented data, which is well-known in the field. This is not an unfair comparison, as we are not altering the network structure; rather, our method revolves around data augmentation. **Therefore, the comparison should focus on the performance before and after the application of data augmentation. Additionally, we would like to clarify the specific performance improvements. The 0.2 increase mentioned by Reviewer S1kV pertains to the ModelNet40 dataset in Table 2, while PointMLP improves by 1.8 on the ScanObjectNN dataset in Table 3.**
> > >
> > > 2) We use the sine function to apply coordinate transformations, which results in different displacements for each point, introducing deformation. This approach is fundamentally different from PointWOLF. We are confused about the argument that translation leads to unfair comparison. **We believe that our comparison is entirely fair, as we are using different methods. PointWOLF requires training and fine-tuning, whereas our approach is simpler, enabling direct end-to-end training without the need for fine-tuning. This is another advantage of our method over PointWOLF.**
> > >
> > > 3) The number of epochs is not a measure of fairness. The epoch count is determined by the specific method used, and each method may differ in this regard. Our comparison is based solely on the results reported in official benchmarks, which is not influenced by epoch numbers. **Furthermore, as shown in the training logs provided in the supplementary materials, we achieved SOTA performance in 240 epochs for DGCNN and 215 epochs for PointNet2-SSG. Our experiments demonstrate that our approach reaches optimal performance within 250 epochs.**

---

> > ### Author Response · Authors · 2024-12-03
> > **Sincerely respond to Reviewer S1kV's comments**
> >
> > Dear PC, SAC, AC , other Reviewers, and Reviewer S1kV,
> >
> > First of all, we have made a detailed reply to the reviewer's initial questions Q1-Q8, and we have not received a positive response until now. The Reviewer S1kV is currently responding to the complaint.
> >
> > We completely disagree with the Reviewer S1kV, and the **evidence** is as follows:
> >
> > 1. Regarding the first response of Reviewer S1kV (they use double the data):
> >
> > 1). It is clear that we are testing a supervised learning task (examples are PointAugment (CVPR2020), PointWOLF (ICCV2021), PCSalMix (ICASSP 2023), etc., which make no use of self-supervised learning), and our paper explicitly incorporates multiple state-of-the-art backbones, as shown in Tables R1 and S1. The Reviewer S1kV has now requested performance on a self-supervised task; however, due to the two-week delay by Reviewer S1kV, it is not feasible to fulfill this request on the final day. As the delay was caused by the Reviewer S1kV, it is not possible for us to complete this task within the remaining time.
> >
> > 2). As the performance on the synthesized dataset ModelNet40 is nearing saturation, the improvement observed is understandably limited. However, on the real-world dataset ScanObjectNN, the performance improvement is clearly evident, as shown in Table S1. **Therefore, the claim that there is insufficient evidence for the effectiveness of our method is not valid. The evaluation of method performance cannot be determined by a single dataset.**
> >
> > **Table S1: 3D shape classification performance in various architectures on PB_T50_RS.**
> > | Model | PointMLP | PointNeXt-S | PointMetaBase-S | SPoTr |
> > |:---|:---:|:---:|:---:|:---:|
> > | | ICLR 2022 | NeurIPS 2022 | CVPR 2023 | CVPR 2023 |
> > | Base | 85.7 | 87.7 | 88.2 | 88.6 |
> > | **+SinPoint(Ours)** | **87.5(+1.8)** | **88.9(+1.2)** | **89.3(+1.1)** | **89.5(+0.9)** |
> >
> > 3) **Once again, we would like to emphasize that our work focuses on data augmentation to expand the dataset. The goal of self-augmentation is to enhance geometric diversity so that when the original data is combined with the augmented data, the model can learn more comprehensive geometric features. This approach is fundamentally different from methods that involve mixing samples.**
> >
> > 4) **We fully understand that reviewing the work on point cloud augmentation in just five hours may pose challenges for Reviewer S1kV, and we appreciate the effort. To assist the Reviewer S1kV, we have provided a brief summary of key points. In the two existing self-augmentation works (PointAugment and PointWOLF), the data was doubled. This is outlined as follows:**
> >
> > **Table S2: Training data under self-augmentation framework.**
> > | Method | Use training data |
> > |:---|:---:|
> > | PointAugment(CVPR2020) | Original + augmented|
> > | PointWOLF(ICCV 2021) | Original + mixture of Original and augmented|
> > | SinPoint(Ours) | Original + augmented|
> >
> > 2. Regarding the second response of Reviewer S1kV:
> >
> > 1) **Regarding the backbone network we use, it incorporates the latest advancements in the field, as detailed in Table S1 and R1.**
> >
> > 2) **While the performance on the synthesized dataset ModelNet40 is approaching saturation, resulting in limited improvements, the performance on the real-world dataset ScanObjectNN shows clear and significant improvement. Again, we direct the Reviewer S1kV to Table S1 for a more detailed comparison. Furthermore, as shown in Table S2, our comparative experiment has been conducted in a completely fair manner.**
> >
> > 3) **We believe that our work is making a significant contribution to the development of point cloud augmentation. As the Reviewer S1kV mentioned in point 4, our code is simple, but isn't it exciting that such simplicity allows us to achieve state-of-the-art (SOTA) performance compared to existing data augmentation methods? This highlights the effectiveness and potential of our approach in advancing the field.**

---

> > ### Author Response · Authors · 2024-12-03
> > **Sincerely respond to Reviewer S1kV's comments**
> >
> > 3. Regarding the third response of Reviewer S1kV:
> >
> > 1) **We kindly remind the Reviewer S1kV to carefully review the data in Table 10. Specifically, when A=0.6, the correct overall accuracy (OA) is 90.189, not 91.189 as may have been mistakenly noted.**
> >
> > 2) **As shown in Table 10, the performance of the model varies with the amplitude A. Specifically, for A=0.2, the overall accuracy (OA) is 88.985; for A=0.4, it is 88.812; for A=0.6, it is 90.189; for A=0.8, it is 89.329; and for A=1.0, it is 88.985. Notably, the performance for all values of A surpasses that of PointWOLF, which achieves a maximum OA of 88.8.**
> >
> > 3) **These results reinforce the robustness of our approach, where even with variations in amplitude, the performance remains competitive and consistently exceeds the performance of PointWOLF.**
> >
> > 4) **In our study, we are specifically investigating the impact of amplitude on model performance. As expected, a very small amplitude leads to poor diversity and reduced performance, while a very large amplitude results in excessive deformation, which negatively affects performance. These findings align perfectly with our theoretical analysis.**
> >
> > 5) **It is clear from Table 10 that A yields the best results, with even the lowest performance (88.812) still achieving state-of-the-art (SOTA) performance. This further demonstrates the effectiveness of our method.**
> >
> > 4. Regarding the fourth response of Reviewer S1kV:
> >
> > 1) **Thanks for your review of our code, It's important for everyone to understand what we do. As you mentioned, our method is simple but highly effective. Upon further review of our paper, you will find that Formulas 4 and 5, Algorithm 1, and Figure 4 all clearly demonstrate the simplicity and effectiveness of our approach. These elements are fully consistent with the implementation in our code.**
> >
> > 2) **While our SinPoint may appear simple, it is grounded in solid theoretical foundations. The existence of homeomorphism ensures that SinPoint enables point cloud augmentation to achieve state-of-the-art (SOTA) performance with just a single sine function. This is not merely a packaging of techniques but rather a theoretical result based on the underlying mathematical principles.**
> >
> > 3) **In contrast, methods like PointWOLF, which rely on the superposition of multiple geometric transformations, lack a theoretical basis for ensuring homeomorphism. This introduces uncertainty, making our method, which guarantees homeomorphic consistency, a more reliable approach.**
> >
> > 4) **Furthermore, our work highlights the importance of point cloud deformation augmentation based on topological consistency—an area that differs fundamentally from the hybrid augmentation techniques commonly used in image processing. We believe this perspective opens new avenues for research in point cloud data augmentation.**
> >
> > 5. Regarding the fifth response of Reviewer S1kV:
> >
> > As the Reviewer S1kV noted, persistent homology is indeed a method for analyzing the topological structure of objects and extracting global topological information, particularly between homeomorphic objects. However, in our approach, we do not require topological analysis. Our focus is on homeomorphic data enhancement, which operates independently of extracting global topological features, as our method does not rely on a full topological analysis of the point cloud. Based on the above facts, we still believe that this is not relevant to our work.
> >
> > **Once again, we kindly request the Reviewer S1kV to carefully review our paper.**
> >
> > **We greatly appreciate the time and effort you dedicate to this review process, and we look forward to receiving your constructive feedback.**
> >
> > Best Regards,
> >
> > The authors of Paper 5358

---

> ### Author Response · Authors · 2024-12-03
> **Sincerely respond to Reviewer S1kV's comments**
>
> 4. Reviewer's Personal Research Experience
>
> 1) We are unable to address the reviewer's personal experience directly. However, regarding the training method, we can clarify that the approach described in the review aligns with the standard parallelization strategy in PyTorch. Specifically, this strategy involves distributing the data across multiple GPUs for training, and it is fully supported by PyTorch. While we use a different training strategy, we would like to emphasize that parallel training is also an option, and our method can support that as well. We are also uncertain about why Reviewer S1kV raised this concern, as our approach is consistent with widely accepted practices.
>
>
> 2) Since the Reviewer S1kV did not provide relevant evidence, we assume that the Reviewer S1kV helpe PointGPT-S increase by 3%.
>
> **Table K2: we hypothesized the experimental results of Reviewer S1kV for comparison**
> | method | OA | dataset | param |
> |:---|:---:|:---:|:---:|
> | pointGPT-S | 86.9 | | 300M |
> | Reviewer S1kV says| 89.9 | 3x | 300M |
> | SPoTr+SinPoint (Our) | 89.5 | 2x | 1.7M |
>
> We would like to respectfully address the fairness concerns raised by Reviewer S1kV. You mentioned improving ScanObjectNN by 3% using a 3x larger dataset. We would like to ask, is it fair to compare this result with the original PointGPT-S, which was trained on a smaller dataset? Additionally, the performance you achieved with a 300M parameter model is impressive, but we would like to point out that we are able to achieve similar performance with a much smaller model of just 1.7M parameters.
>
> 3) Due to the lack of timely feedback from Reviewer S1kV, we were unable to conduct additional experiments within the available time frame. However, we have made substantial progress in the area of supervised learning, which clearly demonstrates the effectiveness of our approach. We have thoroughly verified the performance of our method across various tasks, including point cloud classification, point cloud segmentation, and scene segmentation. We believe these results are sufficient to showcase the strong performance of our method.
>
> 4) We are simply presenting the facts. As shown in Table R1, the following works—SageMix (NeurIPS 2022), WOLFMix (PMLR 2022), PCSalMix (ICASSP 2023), and PointPatchMix (AAAI 2024)—all utilize the same three backbones. Additionally, we have also employed recent backbones such as PointMLP (2022), PointNeXt (2022), PointMetaBase (2023), and SPoTr (2023), which further expands the range of comparisons. Therefore, both the comparison methods and the backbones used are not limited to pre-2019 models. The reviewer's questions still confused us.
>
> 5) We have pioneered the transition in point cloud augmentation from hybrid augmentation to self-deformation augmentation by homeomorphism, which represents a significant advancement in this field. Our self-deformation augmentation clearly outperform all existing hybrid augmentation methods, further demonstrating the effectiveness and potential of our approach.
>
>
> 6) We hope that Reviewer S1kV can carefully review our method, in the our code:
>
> newdata, shift, scale = self.normalize_point_clouds(data)
>
> As we can see, we will normalize the input point cloud and then augment it. And the sine function is a periodic function, which means that the original point cloud must be effectively deformed.
>
> 7) We acknowledge that PointWOLF employs a similar approach (even without the sine transformation, simply applying translation and rotation separately and then feeding them into the network could yield comparable or even better results). However, our method is both simpler and more effective than PointWOLF. We would appreciate it if Reviewer S1kV could provide additional relevant references or studies to support their claim, as we believe our approach offers significant improvements in terms of efficiency and performance.
>
> 5. Minor Issues
>
> As you mentioned, we have already provided a detailed explanation in the appendix. To further assist the reviewers, we will add relevant descriptions in Table 1 for easier reference.
>
> Conclusion
>
> **Every method should be grounded in a solid theoretical foundation. Through our research, we have formally established that if the deformation is not homeomorphic, it results in suboptimal experimental outcomes. The existence of homeomorphism ensures topological consistency, and by applying the sine function to augment point clouds, we are able to maintain this consistency and achieve state-of-the-art performance. We cannot deny the contribution of homeomorphism theory to point cloud augmentation, not only from an engineering perspective but also in terms of its broader theoretical significance.**

---

> > ### Comment · Reviewer_S1kV · 2024-12-03
> >
> > Dear Author,
> >
> > Thank you for your previous responses. I appreciate the effort and work you have put into the rebuttal stage.
> >
> > To be brief, as a professional in the field of point cloud analysis, I quickly noticed that your code is based on the PointMLP GitHub repository, and I am also aware of which dependency files are missing from your code. Based on the configurations you provided, I reproduced your experimental results on two types of GPUs. However, the performance I obtained differs significantly from the results you reported. Although different types of GPUs and configurations may lead to varying outcomes, your reasoning that my training process overfitted seems questionable. I followed the information you provided for training, and since this issue occurred on both types of GPUs, doesn’t this indicate that your code lacks adaptability? I believe the core concern lies in your doubling of the data, which enlarged the training set and caused the model to overfit on it. This appears to be a critical issue, and your code seems to have had a negative effect.
> >
> > At the same time, you misunderstood the parallel operation of data enhancement I mentioned. The parallel operation here is not the parallel operation on the GPU, but you will get triple data after data enhancement is translated, rotated, and randomly dropped. This operation triples the data set. Your method is similar to this operation, and you get two tuples.
> >
> > A minor suggestion: In future work, I suggest validating your data augmentation method in downstream fine-tuning tasks under the current trending self-supervised models, rather than retraining during the pretraining phase. Additionally, it is worth emphasizing that the backbone of PointGPT is approximately 22M in size. When you mentioned that the parameters of PointGPT are around 300M, you likely included other branches from the pretraining process.
> >
> > In conclusion, I thank you again for your efforts and responses. However, you have not fully addressed the core issues I raised. I will maintain my opinions and scores.
> >
> > Best regards,
> > Reviewer S1kV

---

> > > ### Author Response · Authors · 2024-12-04
> > > **Sincerely respond to PC, SAC, AC , other Reviewers, and Reviewer S1kV**
> > >
> > > **We have uploaded an anonymous link in compliance with ICLR guidelines. This link contains the original code, the code submitted to the ICLR supplementary materials, as well as the checkpoints, logs of our training, and specific test times and environment configurations for different GPUs.**
> > >
> > > **The code is available to anyone at https://github.com/Anonymous-code-share/Anonymous. After setting up our environment, we can train it using the following code:**
> > >
> > > **python main.py --model DGCNN --msg test     # on code in original**
> > >
> > > **python main2.py --model DGCNN --msg test   # on code in the supplementary material**
> > >
> > > **We hope this will be helpful to everyone. In the meantime, we welcome anyone who is interested to validate our code.**
> > >
> > > **We look forward to the review of our code by all reviewers as well as PC, SAC, and AC.**
> > >
> > >
> > > Question 2: As a professional in the field of point cloud analysis, I quickly saw that your code is based on the PointMLP github code base, and I also know which dependency files your code is missing. So I am familiar with the field of point clouds. According to the configuration you provided, I quickly ran through your code without making any changes.
> > >
> > > A2: **This issue is related to a missing public library, and our core code remains intact. Additionally, it is incorrect to run the code directly without modifications. Please refer to A1 for further clarification.**
> > >
> > > Question 3: Even though different types of GPUs and configurations will produce different effects, the experimental results I reproduced on two types of GPUs show that the performance you reported is far from what you reported.
> > >
> > > A3: **The incorrect reproduction results are likely due to the failure to modify main.py according to the application details and ablation experiments outlined in the paper. As a result, the reproduction is not aligned with the intended setup. We have provided guidance in A1 to assist with this. We trust this clarifies the issue.**
> > >
> > > Question 4: In addition, for the latest Q2（There is another question. You have been emphasizing local homeomorphism and global homeomorphism in the article, which makes it easy for people to think that local homeomorphism will lead to global homeomorphism. In fact, the two are not equivalent. You did not emphasize this point in the article）,  you asked me to refer to Figure 2, but Figure 2 is just a visualization of the results, without other direct information.
> > >
> > > A4: **We need to clarify that we never stated in the article that local homeomorphism leads to global homeomorphism. This is an extremely imprecise statement.** In **line 161 of the paper**, we explain that **topological consistency can be guaranteed regardless of whether the homeomorphism is global or local**. Furthermore, **in the appendix, we specify that sine transformation ensures local homeomorphism, and under certain special parameters, it can also exhibit global homeomorphism. This guarantees topological consistency in both cases. We hope this clarifies the matter. We believe our theory is well-supported, and by referring to Figure 2 alongside the theoretical discussion, one can better understand both the global and local properties.**
> > >
> > > **We believe that both our theoretical framework and experimental results provide a solid foundation for our work and merit recognition.**
> > >
> > > Best and sincere wishes,
> > >
> > > The authors

---

> ### Author Response · Authors · 2024-12-04
> **Sincerely respond to PC, SAC, AC , other Reviewers, and Reviewer S1kV**
>
> Dear PC, SAC, AC , other Reviewers, and Reviewer S1kV,
>
> Question 1: But after re-reviewing your paper and code, I have a different opinion of your paper. Your code provides a negative impact on your paper.
>
> A1: **We would be happy to assist Reviewer S1kV in reproducing our work. As is well known, for DGCNN, various CDAs need to be incorporated to facilitate experiments. In Table 5, both CDA and dropout must be used correctly. Similarly, for PointNet++, different CDAs are required to support the experiments. We hope this comparison helps clarify the experimental setup.**
>
> **We understand that even simple experiments can fail due to a lack of necessary knowledge. Since the settings vary for each backbone, main.py is not universal.** We have adjusted the code to remove the CDA component, **allowing users to select and add the appropriate CDA as needed. You may need to adjust your strategies accordingly.**
>
> **Since Reviewer S1kV was unable to reproduce our experimental results, we have carefully reviewed our original experiments and related code overnight. We have adjusted all the training parameters without changing any code or parameters, ensuring that the setup is now ready for direct training and testing by others.**
>
> **We have trained on NVIDIA TITAN RTX, RTX 3090, and RTX 4090, and observed no overfitting on the training set across different GPUs. However, the results from Reviewer S1kV show signs of overfitting. Our experiments also confirm that data augmentation helps reduce the risk of overfitting. It is well known that performance may vary slightly across different GPUs. **
>
>
> **Below, we present the training results on code in original:**
>
> We train on NVIDIA TITAN RTX:
>
> The best model trained achieved the following results on the validation set (also used as the test set):
>
> ++++++++++++++++Final results++++++++++++++++
>
> ++  Last Train time: 25 | Last Test time: 1  ++
>
> ++  Best Train loss: 1.225 | Best Test loss: 1.282  ++
>
> ++  Best Train acc_B: 93.477 | Best Test acc_B: 88.666  ++
>
> ++  Best Train acc: 93.945 | Best Test acc: 90.189  ++
>
> ++++++++++++++++++++++++++++++++++++++++
>
> We train on RTX 3090:
>
> The best model trained achieved the following results on the validation set (also used as the test set):
>
> ++++++++++++++++Final results++++++++++++++++
>
> ++  Last Train time: 21 | Last Test time: 1  ++
>
> ++  Best Train loss: 1.218 | Best Test loss: 1.283  ++
>
> ++  Best Train acc_B: 93.862 | Best Test acc_B: 89.024  ++
>
> ++  Best Train acc: 94.444 | Best Test acc: 90.017  ++
>
> ++++++++++++++++++++++++++++++++++++++++
>
> We train on RTX 4090:
>
> The best model trained achieved the following results on the validation set (also used as the test set):
>
> ++++++++++++++++Final results++++++++++++++++
>
> ++  Last Train time: 11 | Last Test time: 1  ++
>
> ++  Best Train loss: 1.223 | Best Test loss: 1.277  ++
>
> ++  Best Train acc_B: 94.287 | Best Test acc_B: 88.906  ++
>
> ++  Best Train acc: 94.705 | Best Test acc: 89.673  ++
>
> ++++++++++++++++++++++++++++++++++++++++
>
> **Below, we present the training results on code in the supplementary material:**
>
> We train on NVIDIA TITAN RTX:
>
> The best model trained achieved the following results on the validation set (also used as the test set):
>
> ++++++++++++++++Final results++++++++++++++++
>
> ++  Last Train time: 24 | Last Test time: 1  ++
>
> ++  Best Train loss: 1.221 | Best Test loss: 1.278  ++
>
> ++  Best Train acc_B: 93.895 | Best Test acc_B: 89.174  ++
>
> ++  Best Train acc: 94.162 | Best Test acc: 90.189  ++
>
> ++++++++++++++++++++++++++++++++++++++++
>
> We train on RTX 3090:
>
> The best model trained achieved the following results on the validation set (also used as the test set):
>
> ++++++++++++++++Final results++++++++++++++++
>
> ++  Last Train time: 18 | Last Test time: 1  ++
>
> ++  Best Train loss: 1.24 | Best Test loss: 1.287  ++
>
> ++  Best Train acc_B: 92.368 | Best Test acc_B: 89.984  ++
>
> ++  Best Train acc: 93.294 | Best Test acc: 90.189  ++
>
> ++++++++++++++++++++++++++++++++++++++++
>
> We train on RTX 4090:
>
> The best model trained achieved the following results on the validation set (also used as the test set):
>
> ++++++++++++++++Final results++++++++++++++++
>
> ++  Last Train time: 12 | Last Test time: 1  ++
>
> ++  Best Train loss: 1.241 | Best Test loss: 1.286  ++
>
> ++  Best Train acc_B: 92.675 | Best Test acc_B: 89.085  ++
>
> ++  Best Train acc: 93.381 | Best Test acc: 89.501  ++
>
> ++++++++++++++++++++++++++++++++++++++++

---

### Note · Authors · 2025-01-23

I have read and agree with the venue's withdrawal policy on behalf of myself and my co-authors.